# Complex Wavelet Mutual Information Loss: A Multi-Scale Loss Function for Semantic Segmentation

**Renhao Lu** [1]

## Abstract

Recent advancements in deep neural networks have significantly enhanced the performance of semantic segmentation. However, class imbalance and instance imbalance remain persistent challenges, where smaller instances and thin boundaries are often overshadowed by larger structures. To address the multiscale nature of segmented objects, various models have incorporated mechanisms such as spatial attention and feature pyramid networks. Despite these advancements, most loss functions are still primarily pixel-wise, while regional and boundary-focused loss functions often incur high computational costs or are restricted to small-scale regions. To address this limitation, we propose the complex wavelet mutual information (CWMI) loss, a novel loss function that leverages mutual information from subband images decomposed by a complex steerable pyramid. The complex steerable pyramid captures features across multiple orientations and preserves structural similarity across scales. Meanwhile, mutual information is well-suited to capturing high-dimensional directional features and offers greater noise robustness. Extensive experiments on diverse segmentation datasets demonstrate that CWMI loss achieves significant improvements in both pixel-wise accuracy and topological metrics compared to state-of-the-art methods, while introducing minimal computational overhead. Our code is available at https://github.com/lurenhaothu/CWMI

## 1. Introduction

Semantic segmentation, the process of partitioning an image into regions associated with semantic labels, plays a crucial role in applications ranging from autonomous driving to biomedical imaging. Despite significant progress driven by deep neural networks such as U-Net (Ronneberger et al., 2015) and fully convolutional networks (Long et al., 2015), challenges persist. Class imbalance, where dominant classes overshadow smaller ones, and instance imbalance, where small-scale structures are frequently ignored, remain major obstacles (Jiang et al., 2024; Kofler et al., 2023). Addressing these imbalances requires not only pixel-wise accuracy, but also the preservation of structural similarity, a property vital for ensuring spatial coherence and topological integrity in segmented outputs (Wang et al., 2004).

Although the advancements in feature extraction, such as feature pyramid networks (Lin et al., 2017) and attention mechanisms (Woo et al., 2018; Chen et al., 2021; Islam et al., 2020), have enabled models to capture multiscale contextual information, most loss functions still focus on pixel-wise optimization (Azad et al., 2023). Region-based loss functions, such as Dice loss (Milletari et al., 2016) and Tversky loss (Salehi et al., 2017), are not subject to class imbalance, but smaller instances within the same class are still easy to be overshadowed, which can be critical for preserving regional and boundary details. Zhao et al. developed Regional Mutual Information (RMI) loss, which captures statistical relationships over regions. However, RMI is constrained within a relatively small region ($3 \times 3$ pixels) to avoid high computational overhead (Zhao et al., 2019). Thus, a loss function that balances computational efficiency with the ability to model structural and regional dependencies at larger scales is highly desirable.

Addressing these challenges from a frequency-domain perspective provides an innovative pathway. Patterns of varying scales in images are inherently tied to their frequency components: large-scale structures correspond to low frequencies, while finer details correspond to high frequencies. Wavelet transforms are uniquely suited for this multiscale decomposition, as they preserve both spatial and frequency information (Mallat, 1989). Among these, the steerable pyramid, proposed by (Simoncelli et al., 1992), leverages

[1]Meinig School of Biomedical Engineering, Cornell University, Ithaca, NY, USA. Correspondence to: Renhao Lu <rl839@cornell.edu>.

*Proceedings of the 42$^{nd}$ International Conference on Machine Learning*, Vancouver, Canada. PMLR 267, 2025. Copyright 2025 by the author(s).

steerable filters for redundant wavelet decomposition, enabling multiscale and multi-orientation feature extraction. Its extension, the complex steerable pyramid, further enhances this framework by using complex numbers to explicitly represent local phase information, allowing for robust extraction of structural details across scales and orientations (Portilla & Simoncelli, 2000). These properties make it a powerful tool for segmentation tasks where structural similarity is paramount.

In this paper, we introduce Complex Wavelet Mutual Information (CWMI) loss, a novel loss function that leverages the complex steerable pyramid for efficient multiscale structural information extraction. By combining the robust multiscale decomposition capabilities of the complex steerable pyramid with the statistical power of mutual information, CWMI loss explicitly incorporates local phase, orientation, and structural features into the loss calculation. This approach ensures structural coherence and boundary preservation while maintaining computational efficiency, making it well-suited for segmentation tasks with significant class and instance imbalances.

Our contributions are summarized as follows:

- We propose CWMI loss, which can maximize the mutual information in the domain of complex steerable pyramid decompositions. Such a strategy can enhance multiscale structural features for semantic segmentation, especially for tasks with significant class and instance imbalances.

- We demonstrated the superiority of CWMI with extensive experiments on four public segmentation datasets: SNEMI3D (neurite segmentation in electron microscopy slices), GlaS (gland segmentation in H&E slices), DRIVE (retinal vessel segmentation in fundus images), and MASS ROAD (road segmentation from aerial imagery). Compared with 11 state-of-art (SOTA) loss functions, CWMI showed better performance on both pixel-wise metrics and topological metrics, while introducing minimal computational overhead.

## 2. Related Work

Semantic segmentation has seen tremendous advancements through deep learning architectures, with U-Net and its variants becoming a cornerstone of this field. The U-Net model, proposed by (Ronneberger et al., 2015), utilizes a symmetric encoder-decoder architecture with skip connections to preserve spatial details while capturing global context. Its success has inspired numerous adaptations in its convolutional blocks (Huang et al., 2019; Diakogiannis et al., 2020) and skip connections (Zhou et al., 2018; Chen et al., 2021). Inspired by the transformer model, the attention mechanism has also been incorporated into the U-Net structure, which

has shown significant performance enhancement, as in Attention U-Net (Islam et al., 2020), TransUNet (Chen et al., 2021), Vision Mamba UNet (Ruan et al., 2024), etc. In this study, we compare our proposed CWMI loss using U-Net and Attention U-Net to test the generalization and superiority of CWMI, while the general idea is adaptable to other architectures as well.

### 2.1. Loss Functions for Semantic Segmentation

While the architectural advancements in segmentation models have been significant, the performance of these networks is highly influenced by the design of the loss functions. Pixel-wise cross entropy loss (CE) minimizes the log likelihood of the prediction error but is significantly prone to class imbalance. To address this issue, class balanced cross entropy (BCE), proposed by (Long et al., 2015), employs higher weights for classes with fewer pixels. Focal loss assigns higher weights to misclassified pixels with high probabilities (Ross & Dollár, 2017). Region-based losses, including Dice loss (Milletari et al., 2016), Tversky loss (Salehi et al., 2017), and Jaccard loss (Rahman & Wang, 2016), inherently handle class imbalance but fail to address instance imbalance within the same class, such as thin boundaries and small objects.

To tackle instance imbalance, weighted loss functions have been proposed. In the original U-Net paper (Ronneberger et al., 2015), the weighted cross entropy (WCE) was introduced, employing a distance-based weight map to emphasize thin boundaries between objects. However, WCE assigns weights only to boundary pixels, neglecting object pixels. The adaptive boundary weighted (ABW) loss (Liu et al., 2022) extends this approach by applying distance-based weights to both boundary and object pixels, while the Skea-Topo loss further improved the weight map based on boundary and object skeletons (Liu et al., 2024). Despite their contributions, weighted losses suffer from two major limitations: (1) the weight maps are precomputed and fixed, failing to adapt to errors during training, and (2) they often generate thicker boundaries, which preserve topology but compromise metrics like Dice score and mIoU, as observed in our qualitative results.

Several methods address instance imbalance dynamically during training, but at the cost of computational efficiency. Topology-based approaches, such as persistent homology methods (Stucki et al., 2023; Oner et al., 2023), describe image topologies and identify critical pixels but are computationally expensive, with cubic complexity to image size. The clDice loss (Shit et al., 2021) employs a soft skeletonization algorithm to detect topological errors, primarily focusing on thin-boundary objects like retinal blood vessels. Similarly, Boundary Loss (Kervadec et al., 2019) and Hausdorff Distance Loss (Karimi & Salcudean, 2019) refine boundaries but incur significant computational overhead.

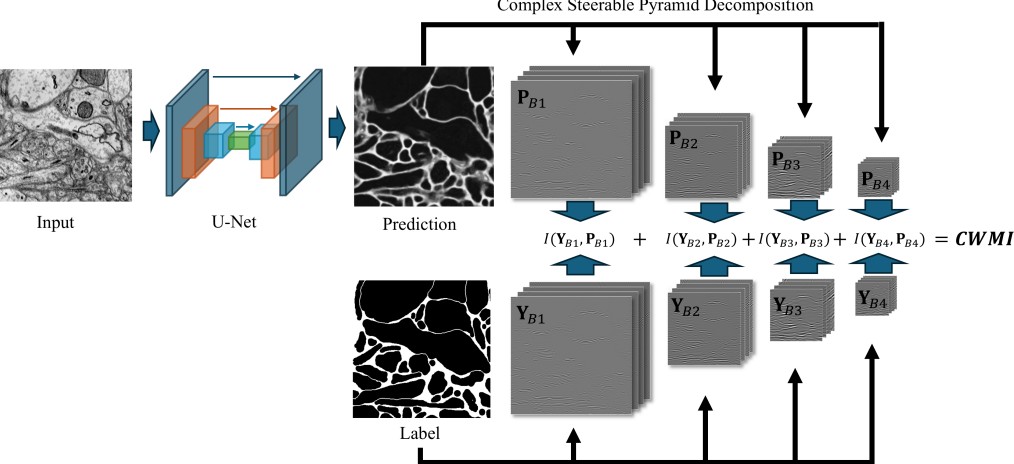

*Figure 1.* Illustration of the proposed Complex Wavelet Mutual Information (CWMI) Loss. The prediction and label images are decomposed using a complex steerable pyramid, which generates subbands at different scales and orientations. Mutual information is calculated for each corresponding pair of subbands, and the CWMI is computed as the sum of these mutual information values. $\mathbf{Y}_{B_n}, \mathbf{P}_{B_n}$: complex steerable decomposition of label and prediction image at level $n$; $I(\mathbf{Y}_{B_n}, \mathbf{Y}_{B_n})$: mutual information between $\mathbf{Y}_{B_n}$ and $\mathbf{P}_{B_n}$

Region Mutual Information (RMI) loss (Zhao et al., 2019) captures pixel interdependencies over regions, but struggles with scalability for large-scale regional analysis due to the high computation cost. These losses either prioritize small regions at the expense of global accuracy or require extensive computational resources, necessitating more efficient and balanced approaches.

### 2.2. Wavelet-Based Loss Functions

Wavelet-based metrics were first introduced as Complex Wavelet Structural Similarity (CW-SSIM) (Sampat et al., 2009), known for their robustness to small translations and rotations. In the deep learning era, wavelet-based methods have been employed in loss functions, leveraging their ability to analyze multiscale and multiresolution features. These methods have shown promise in tasks like sketch-to-image translation (Kim & Cho, 2023), image super-resolution (Korkmaz & Tekalp, 2024), image dehazing (Yang et al., 2020), and material analysis (Prantl et al., 2022). However, to the best of our knowledge, wavelet-based loss functions have yet to be explored in semantic segmentation.

Existing wavelet-based loss functions typically rely on $L_1$ (Zhu et al., 2021; Korkmaz & Tekalp, 2024; Prantl et al., 2022) or $L_2$ distances (Kim & Cho, 2023), or structural similarity (SSIM) (Yang et al., 2020) in the decomposed domain. While effective, these methods are less suited for handling high-dimensional data with complex directional features, as in wavelet transforms, and may be vulnerable to noise. The proposed CWMI loss leverages mutual information between wavelet-based subband images, effectively capturing multiscale dependencies. As demonstrated in later ablation tests, CWMI outperforms traditional metrics like $L_1$, $L_2$,

and SSIM, offering superior segmentation performance and robustness.

## 3. Methods

To compute the proposed Complex Wavelet Mutual Information (CWMI) loss, the prediction and ground truth label matrices are first decomposed into subbands using the complex steerable pyramid. Mutual information (MI) is then calculated for each subband. The CWMI loss is defined as the sum of the MI values across all subbands, as illustrated in Figure 1.

### 3.1. Complex Wavelet Decomposition

**Wavelet Transform**   Wavelet transforms are widely used for multiscale analysis, enabling decomposition of an image into frequency subbands while preserving spatial information. Unlike traditional Fourier transforms, which analyze global frequency components, wavelet transforms provide a localized frequency representation, making them well-suited for tasks involving spatially-varying structures such as semantic segmentation. For an image $I(x, y)$, the wavelet transform is defined as:

$$W_\psi(s, t) = \int \int I(x, y)\psi_{s,t}(x, y)\, dx\, dy \qquad (1)$$

where $\psi_{s,t}(x, y)$ is a scaled and translated version of the mother wavelet function $\psi(x, y)$, with $s$ controlling the scale and $t$ the translation.

For discrete signals, the Discrete Wavelet Transform (DWT) decomposes an image into progressively lower resolution subbands using filter banks. However, traditional wavelet

decompositions suffer from limited orientation selectivity, capturing only fixed horizontal, vertical, and diagonal directions.

**Steerable Pyramid** To overcome these limitations, steerable pyramid extends the wavelet framework by introducing orientation-sensitive band-pass filters, significantly enhancing orientation selectivity. Unlike DWT, which provides a non-redundant representation, the steerable pyramid offers a flexible, redundant image representation, facilitating improved multiscale analysis. This decomposition is achieved through the iterative application of steerable band-pass filters followed by downsampling. In the frequency domain, the band-pass filter for the $k_{th}$ orientation is expressed in polar coordinates $(r, \theta)$ as:

$$B_k(r, \theta) = H(r)G_k(\theta), \quad k \in [1, K], \qquad (2)$$

where $H(r)$ and $G_k(\theta)$ represent the radial and angular components, respectively:

$$H(r) = \begin{cases} \cos\left(\frac{\pi}{2} \log_2\left(\frac{2r}{\pi}\right)\right), & \frac{\pi}{4} < r < \frac{\pi}{2}, \\ 1, & r \leq \frac{\pi}{2}, \\ 0, & r \geq \frac{\pi}{4}. \end{cases} \qquad (3)$$

$$G_k(\theta) = \alpha_k \left| \cos\left(\theta - \frac{\pi k}{K}\right) \right|^{K-1}, \qquad (4)$$

where $K$ is the number of orientations and $\alpha_k = 2^{k-1} \frac{(K-1)!}{\sqrt{K[2(K-1)]!}}$. Figure 2A provides an illustration of the band filter with $K = 4$.

With total recursive levels $N$, an image $I$ can be decomposed as:

$$\mathbf{I} \rightarrow \begin{vmatrix} \mathbf{I}_{H_0} \in \mathbf{R}^{H_0 \times W_0}, \\ \mathbf{I}_{B_1} \in \mathbf{R}^{K \times H_0 \times W_0}, \\ \mathbf{I}_{B_2} \in \mathbf{R}^{K \times H_1 \times W_1}, \\ \dots \\ \mathbf{I}_{B_N} \in \mathbf{R}^{K \times H_{N-1} \times W_{N-1}}, \\ \mathbf{I}_{L_0} \in \mathbf{R}^{H_{N-1} \times W_{N-1}}, \end{vmatrix} \qquad (5)$$

where $\mathbf{I}_{H_0}$ and $\mathbf{I}_{L_0}$ are high-frequency and low-frequency residues, and $\mathbf{I}_{B_n}$ represents subband images at level $n$ with $k$-th direction concatenated in the first dimension. Figure 2B shows an example of the decomposed output of an input image $\mathbf{I}$.

**Complex Steerable Pyramid** Although the steerable pyramid effectively captures amplitude information across multiple orientations, it lacks the ability to extract local phase information, which is crucial for encoding structural features such as edges and corners (Canny, 1986). To address this limitation, (Portilla & Simoncelli, 2000) introduced the complex steerable pyramid, which extends the conventional steerable pyramid by converting its decomposed images into their analytic signal representation. In

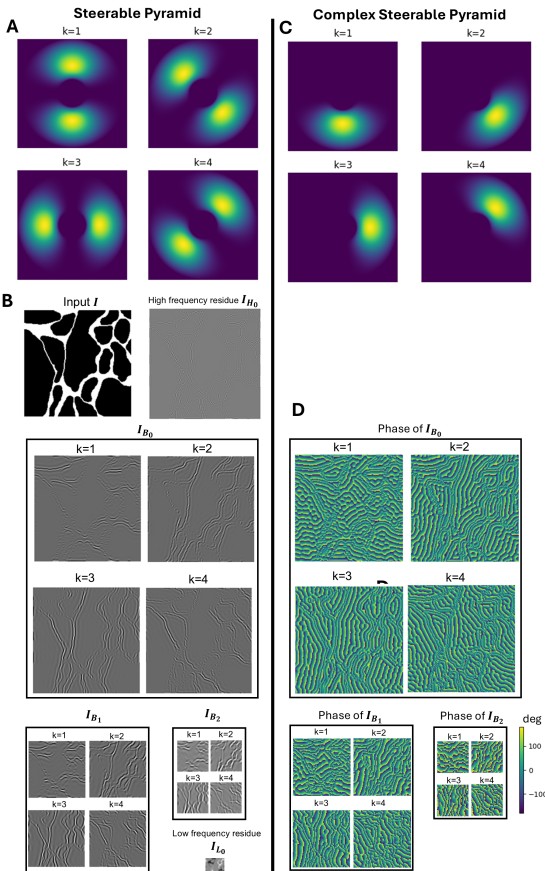

*Figure 2.* Steerable pyramid and complex steerable pyramid. (A) Orientation-selective band-pass filters of the steerable pyramid. (B) Example decomposition using a steerable pyramid with N=3, K=4. (C) Band-pass filters of the complex steerable pyramid, where negative frequency components are discarded. (D) Phase representation of the complex steerable pyramid output, with the real part identical to that of the steerable pyramid.

this formulation, the real part remains unchanged, while the imaginary part is obtained via the Hilbert transform of the real component. In the Fourier domain, this transformation is equivalent to discarding negative frequency components, as illustrated in Figure 2C.

For the complex steerable pyramid, the angular component $G_k(\theta)$ is modified as:

$$\tilde{G}_k(\theta) = \begin{cases} 2\alpha_k \left[\cos\left(\theta - \frac{\pi k}{K}\right)\right]^{K-1}, & \left|\theta - \frac{\pi k}{K}\right| < \frac{\pi}{2}, \\ 0, & \text{otherwise.} \end{cases} \qquad (6)$$

This modification enables image decomposition into complex subbands, where phase information encodes critical structural features such as edges and corners (Figure 2D), while amplitude represents feature strength.

## 3.2. Mutual Information in Complex Wavelet Domain

According to Equation 5, the ground truth $\mathbf{Y}$ and the prediction $\mathbf{P}$ are decomposed into subbands $\mathbf{Y}_{B_n}$ and $\mathbf{P}_{B_n} \in \mathbf{R}^{K \times H_{n-1} \times W_{n-1}}$ for each level $n \in [1, N]$. For each pixel $(x, y)$ at level $n$, the $K$-directional features are treated as $K$-dimensional random variables.

For mutual information approximation, several studies, such as MINE (Belghazi et al., 2018) and Deep InfoMax (Hjelm et al., 2018), employ neural network-based estimators to produce tight lower bounds. While these methods are theoretically well-founded, they introduce additional training overhead, which increases algorithmic complexity and may hinder the efficiency of the loss function. Therefore, in this work, we adopt the mutual information estimation approach proposed by (Zhao et al., 2019):

$$I_l(\mathbf{Y}_{B_n}; \mathbf{P}_{B_n}) \approx -\frac{1}{2} \log \det(\mathbf{M}_n), \qquad (7)$$

$$\mathbf{M}_n = \mathbf{\Sigma}_{\mathbf{Y}_{B_n}} - \mathrm{Cov}(\mathbf{Y}_{B_n}, \mathbf{P}_{B_n})(\mathbf{\Sigma}_{\mathbf{P}_{B_n}}^{-1})^T \mathrm{Cov}(\mathbf{Y}_{B_n}, \mathbf{P}_{B_n})^T \qquad (8)$$

where $\mathbf{\Sigma}_{\mathbf{Y}_{B_n}}$ and $\mathbf{\Sigma}_{\mathbf{P}_{B_n}}$ are covariance matrices, and $\mathrm{Cov}(\mathbf{Y}_{B_n}, \mathbf{P}_{B_n})$ is the cross-covariance.

When the ground truth and predictions are decomposed into complex subbands, this equation is extended with Hermitian transpose $H$:

$$\tilde{\mathbf{M}}_n = \mathbf{\Sigma}_{\mathbf{Y}_{B_n}} - \mathrm{Cov}(\mathbf{Y}_{B_n}, \mathbf{P}_{B_n})(\mathbf{\Sigma}_{\mathbf{P}_{B_n}}^{-1})^H \mathrm{Cov}(\mathbf{Y}_{B_n}, \mathbf{P}_{B_n})^H \qquad (9)$$

where the covariance and cross-covariance are calculated as:

$$\mathbf{\Sigma}_{\mathbf{Y}_{B_n}} = E[(\mathbf{Y}_{B_n} - E[\mathbf{Y}_{B_n}])(\mathbf{Y}_{B_n} - E[\mathbf{Y}_{B_n}])^H] \quad (10)$$

$$\mathbf{\Sigma}_{\mathbf{P}_{B_n}} = E[(\mathbf{P}_{B_n} - E[\mathbf{P}_{B_n}])(\mathbf{P}_{B_n} - E[\mathbf{P}_{B_n}])^H] \quad (11)$$

$$\mathrm{Cov}(\mathbf{Y}_{B_n}, \mathbf{P}_{B_n}) = E[(\mathbf{Y}_{B_n} - E[\mathbf{Y}_{B_n}])(\mathbf{P}_{B_n} - E[\mathbf{P}_{B_n}])^H] \quad (12)$$

Finally, the CWMI loss is computed as the sum of MI across all levels, which combines with cross entropy loss to integrate pixel-wise loss:

$$CWMI(\mathbf{Y}, \mathbf{P}) = (1-\lambda) \sum_{n=1}^{N} -I_l(\mathbf{Y}_{B_n}; \mathbf{P}_{B_n}) + \lambda L_{ce}(\mathbf{Y}, \mathbf{P}). \qquad (13)$$

## 4. Experiment

### 4.1. Experimental Setup

**Base Models** To evaluate the effectiveness of the CWMI loss, we employed U-Net (Ronneberger et al., 2015) and Attention U-Net (Oktay et al., 2018) as baseline architectures.

This selection enables an assessment of CWMI's generalization ability across both fully convolutional and attention-enhanced models. Additionally, to examine CWMI's compatibility with recently proposed Mamba-based architectures (Yue & Li, 2024), we incorporated the Vision Mamba U-Net (VMUNet) (Ruan et al., 2024) into our experiments.

**Datasets** We tested CWMI on three public segmentation datasets, all characterized by class and instance imbalance: (1) SNEMI3D, a neurite segmentation dataset containing 100 $1024 \times 1024$ grayscale images from electron microscopy slices (Arganda-Carreras et al., 2013); (2) GlaS, a gland segmentation dataset with 165 RGB images of varying sizes from histological images of colorectal cancer samples (Sirinukunwattana et al., 2017); (3) DRIVE, a retinal vessel segmentation dataset comprising 40 $584 \times 565$ RGB images from fundus photographs (Staal et al., 2004); and (4) the Massachusetts Roads dataset (MASS ROAD), a road segmentation dataset with 1171 $1500 \times 1500$ RGB images from aerial imagery (Mnih, 2013). We choose a subset of 120 images (ignoring images without a network of roads). For all datasets, at least two repeated three-fold cross-validations were used to ensure robust evaluation.

**Baselines and Implementation Details** We compared CWMI against 11 state-of-the-art (SOTA) loss functions, including pixel-wise loss functions (e.g., cross entropy, BCE (Long et al., 2015), Focal loss (Ross & Dollár, 2017)), Sensitive loss (Tang et al., 2025), region-based loss functions (e.g., Dice loss (Milletari et al., 2016), Tversky loss (Salehi et al., 2017), Jaccard loss (Rahman & Wang, 2016)), and structural/topological loss functions (e.g., WCE (Ronneberger et al., 2015), ABW loss (Liu et al., 2022), Skea-topo loss (Liu et al., 2024), RMI loss (Zhao et al., 2019), clDice loss (Shit et al., 2021)). Hyperparameters for each baseline were tuned via grid search, with Tversky loss ($\alpha = .5, \beta = .5$) and Focal loss ($\gamma = 2.5$) as examples.

We utilized steerable pyramids with four decomposition levels and four orientations in all experiments. $N = 4$ and $K = 4$ of the steerable pyramid decomposition and a regularization parameter of $\lambda = 0.1$ were determined by the ablation experiments, and applied in all CWMI experiments. For the implementation of Equation 9, although PyTorch supports complex matrix calculations, our experiments indicated that its efficiency remains suboptimal. Consequently, we computed using their real representations, which are mathematically equivalent (Golub & Van Loan, 2013). To assess the significance of the complex steerable pyramid, we compared CWMI with a real-number-only variant (CWMI-Real), implemented according to Equations 4 and 8. Adam optimizer with a StepLR scheduler (initial learning rate $1 \times 10^{-4}$, decay rate 0.8, step size 10) was used. U-Net and Attention U-Net models were trained for 50 epochs; VMUNet models were trained for 100 epochs due to their slower convergence. All models were trained with a batch

**SENMI3D**

| Methods | UNet | | | | | AttenUNet | | | | |
|---|---|---|---|---|---|---|---|---|---|---|
| | mIoU↑ | mDice↑ | VI↓ | ARI↑ | HD↓ | mIoU↑ | mDice↑ | VI↓ | ARI↑ | HD↓ |
| CE | $.753_{\pm.005}$ | $.851_{\pm.004}$ | $2.082_{\pm.270}$ | $.500_{\pm.039}$ | $1.143_{\pm.176}$ | $.751_{\pm.009}$ | $.850_{\pm.007}$ | $1.914_{\pm.361}$ | $.525_{\pm.056}$ | $1.178_{\pm.261}$ |
| BCE | $.736_{\pm.006}$ | $.841_{\pm.004}$ | $1.756_{\pm.156}$ | $.561_{\pm.020}$ | $1.413_{\pm.088}$ | $.735_{\pm.010}$ | $.840_{\pm.007}$ | $1.835_{\pm.150}$ | $.553_{\pm.023}$ | $1.417_{\pm.224}$ |
| Dice | $.767_{\pm.004}$ | $.862_{\pm.003}$ | $1.406_{\pm.085}$ | $.604_{\pm.017}$ | $.919_{\pm.154}$ | $.768_{\pm.005}$ | $.862_{\pm.004}$ | $1.507_{\pm.086}$ | $.583_{\pm.016}$ | $\mathbf{.804_{\pm.072}}$ |
| Focal | $.728_{\pm.007}$ | $.835_{\pm.005}$ | $1.751_{\pm.078}$ | $.556_{\pm.016}$ | $1.725_{\pm.225}$ | $.725_{\pm.005}$ | $.833_{\pm.003}$ | $1.911_{\pm.189}$ | $.540_{\pm.019}$ | $1.803_{\pm.107}$ |
| Jaccard | $.766_{\pm.004}$ | $.861_{\pm.003}$ | $1.357_{\pm.114}$ | $.614_{\pm.019}$ | $.904_{\pm.087}$ | $.762_{\pm.003}$ | $.858_{\pm.002}$ | $1.360_{\pm.079}$ | $.609_{\pm.007}$ | $1.000_{\pm.125}$ |
| Tversky | $.765_{\pm.004}$ | $.861_{\pm.003}$ | $1.328_{\pm.077}$ | $.617_{\pm.017}$ | $.990_{\pm.183}$ | $.765_{\pm.003}$ | $.860_{\pm.002}$ | $1.303_{\pm.057}$ | $.621_{\pm.009}$ | $.969_{\pm.101}$ |
| WCE | $.714_{\pm.005}$ | $.825_{\pm.003}$ | $1.987_{\pm.398}$ | $.525_{\pm.042}$ | $1.575_{\pm.250}$ | $.713_{\pm.005}$ | $.825_{\pm.004}$ | $1.818_{\pm.034}$ | $.543_{\pm.009}$ | $1.613_{\pm.192}$ |
| ABW | $.605_{\pm.066}$ | $.739_{\pm.057}$ | $3.637_{\pm1.703}$ | $.299_{\pm.187}$ | $4.605_{\pm2.926}$ | $.616_{\pm.072}$ | $.749_{\pm.063}$ | $3.077_{\pm1.303}$ | $.366_{\pm.142}$ | $4.045_{\pm3.220}$ |
| Skea-topo | $.572_{\pm.149}$ | $.671_{\pm.197}$ | $3.822_{\pm2.560}$ | $.300_{\pm.259}$ | $1.618_{\pm.207}$ | $.602_{\pm.095}$ | $.729_{\pm.096}$ | $3.759_{\pm2.324}$ | $.306_{\pm.243}$ | $4.889_{\pm5.928}$ |
| RMI | $.764_{\pm.006}$ | $.859_{\pm.005}$ | $1.443_{\pm.207}$ | $.587_{\pm.037}$ | $.943_{\pm.130}$ | $.764_{\pm.008}$ | $.859_{\pm.006}$ | $1.404_{\pm.124}$ | $.585_{\pm.028}$ | $.958_{\pm.170}$ |
| clDice | $.706_{\pm.014}$ | $.819_{\pm.011}$ | $2.049_{\pm.351}$ | $.502_{\pm.052}$ | $1.489_{\pm.218}$ | $.701_{\pm.022}$ | $.815_{\pm.016}$ | $1.887_{\pm.161}$ | $.518_{\pm.034}$ | $1.720_{\pm.244}$ |
| Sensitive | $.763_{\pm.006}$ | $.858_{\pm.005}$ | $1.543_{\pm.069}$ | $.576_{\pm.023}$ | $.948_{\pm.143}$ | $.757_{\pm.006}$ | $.854_{\pm.004}$ | $1.822_{\pm.313}$ | $.527_{\pm.049}$ | $1.016_{\pm.139}$ |
| CWMI-Real | $.776_{\pm.004}$** | $.867_{\pm.003}$** | $1.205_{\pm.059}$** | $.634_{\pm.013}$* | $.807_{\pm.117}$* | $.775_{\pm.004}$** | $.867_{\pm.003}$** | $1.193_{\pm.070}$** | $.634_{\pm.014}$* | $.824_{\pm.103}$ |
| CWMI | $\mathbf{.778_{\pm.004}}$*** | $\mathbf{.869_{\pm.003}}$*** | $\mathbf{1.162_{\pm.068}}$*** | $\mathbf{.638_{\pm.015}}$* | $\mathbf{.739_{\pm.095}}$** | $\mathbf{.777_{\pm.006}}$** | $\mathbf{.868_{\pm.005}}$** | $\mathbf{1.162_{\pm.085}}$** | $\mathbf{.639_{\pm.019}}$* | $.807_{\pm.069}$ |

**GlaS**

| Methods | UNet | | | | | AttenUNet | | | | |
|---|---|---|---|---|---|---|---|---|---|---|
| | mIoU↑ | mDice↑ | VI↓ | ARI↑ | HD↓ | mIoU↑ | mDice↑ | VI↓ | ARI↑ | HD↓ |
| CE | $.637_{\pm.204}$ | $.724_{\pm.192}$ | $.933_{\pm.043}$ | $.400_{\pm.333}$ | $6.633_{\pm4.654}$ | $.640_{\pm.209}$ | $.725_{\pm.195}$ | $.906_{\pm.107}$ | $.406_{\pm.327}$ | $7.914_{\pm6.637}$ |
| BCE | $.614_{\pm.226}$ | $.702_{\pm.213}$ | $1.032_{\pm.152}$ | $.355_{\pm.355}$ | $4.165_{\pm1.929}$ | $.619_{\pm.222}$ | $.708_{\pm.208}$ | $.986_{\pm.139}$ | $.365_{\pm.358}$ | $4.354_{\pm1.869}$ |
| Dice | $.632_{\pm.201}$ | $.719_{\pm.189}$ | $.887_{\pm.080}$ | $.396_{\pm.322}$ | $7.004_{\pm4.670}$ | $.636_{\pm.197}$ | $.723_{\pm.187}$ | $.954_{\pm.056}$ | $.396_{\pm.322}$ | $7.335_{\pm4.843}$ |
| Focal | $.581_{\pm.253}$ | $.672_{\pm.241}$ | $1.093_{\pm.103}$ | $.330_{\pm.356}$ | $4.447_{\pm2.023}$ | $.581_{\pm.252}$ | $.672_{\pm.240}$ | $1.043_{\pm.109}$ | $.336_{\pm.369}$ | $5.040_{\pm2.802}$ |
| Jaccard | $.637_{\pm.205}$ | $.723_{\pm.193}$ | $.947_{\pm.076}$ | $.397_{\pm.335}$ | $6.979_{\pm4.511}$ | $.638_{\pm.202}$ | $.725_{\pm.190}$ | $.888_{\pm.076}$ | $.406_{\pm.331}$ | $6.758_{\pm4.431}$ |
| Tversky | $.634_{\pm.205}$ | $.722_{\pm.193}$ | $.953_{\pm.031}$ | $.394_{\pm.332}$ | $6.705_{\pm4.398}$ | $.627_{\pm.195}$ | $.717_{\pm.186}$ | $.924_{\pm.102}$ | $.384_{\pm.314}$ | $7.521_{\pm4.493}$ |
| WCE | $.829_{\pm.010}$ | $.902_{\pm.007}$ | $.832_{\pm.075}$ | $.721_{\pm.008}$ | $2.264_{\pm.210}$ | $.823_{\pm.016}$ | $.898_{\pm.011}$ | $.873_{\pm.097}$ | $.705_{\pm.023}$ | $2.590_{\pm.602}$ |
| ABW | $.760_{\pm.012}$ | $.857_{\pm.008}$ | $1.468_{\pm.079}$ | $.619_{\pm.027}$ | $3.527_{\pm.508}$ | $.762_{\pm.019}$ | $.858_{\pm.013}$ | $1.472_{\pm.103}$ | $.614_{\pm.037}$ | $3.623_{\pm.887}$ |
| Skea-topo | $.788_{\pm.008}$ | $.876_{\pm.005}$ | $1.259_{\pm.042}$ | $.658_{\pm.002}$ | $2.985_{\pm.765}$ | $.784_{\pm.016}$ | $.873_{\pm.010}$ | $1.299_{\pm.086}$ | $.645_{\pm.022}$ | $3.363_{\pm.609}$ |
| RMI | $.839_{\pm.011}$ | $.907_{\pm.007}$ | $.820_{\pm.082}$ | $.726_{\pm.024}$ | $2.644_{\pm.372}$ | $.835_{\pm.015}$ | $.905_{\pm.010}$ | $.822_{\pm.083}$ | $.720_{\pm.039}$ | $2.701_{\pm.366}$ |
| clDice | $.816_{\pm.017}$ | $.894_{\pm.012}$ | $.920_{\pm.107}$ | $.698_{\pm.026}$ | $2.809_{\pm.476}$ | $.800_{\pm.013}$ | $.884_{\pm.009}$ | $1.004_{\pm.113}$ | $.684_{\pm.026}$ | $3.146_{\pm.286}$ |
| Sensitive | $.821_{\pm.014}$ | $.897_{\pm.009}$ | $.941_{\pm.101}$ | $.694_{\pm.033}$ | $2.916_{\pm.394}$ | $.824_{\pm.018}$ | $.899_{\pm.011}$ | $.880_{\pm.108}$ | $.702_{\pm.041}$ | $2.757_{\pm.398}$ |
| CWMI-Real | $.838_{\pm.014}$ | $.907_{\pm.010}$ | $.798_{\pm.068}$ | $.727_{\pm.027}$ | $2.813_{\pm.653}$ | $.842_{\pm.022}$ | $.909_{\pm.015}$ | $.788_{\pm.063}$ | $.724_{\pm.038}$ | $2.758_{\pm.764}$ |
| CWMI | $\mathbf{.843_{\pm.016}}$ | $\mathbf{.910_{\pm.011}}$ | $\mathbf{.761_{\pm.081}}$* | $\mathbf{.735_{\pm.026}}$ | $\mathbf{2.569_{\pm.466}}$ | $\mathbf{.844_{\pm.007}}$ | $\mathbf{.911_{\pm.005}}$ | $\mathbf{.755_{\pm.106}}$ | $\mathbf{.737_{\pm.015}}$ | $\mathbf{2.569_{\pm.547}}$ |

**DRIVE**

| Methods | UNet | | | | | AttenUNet | | | | |
|---|---|---|---|---|---|---|---|---|---|---|
| | mIoU↑ | mDice↑ | VI↓ | ARI↑ | HD↓ | mIoU↑ | mDice↑ | VI↓ | ARI↑ | HD↓ |
| CE | $.770_{\pm.014}$ | $.856_{\pm.011}$ | $1.379_{\pm.136}$ | $.406_{\pm.059}$ | $2.344_{\pm.398}$ | $.757_{\pm.017}$ | $.845_{\pm.014}$ | $1.436_{\pm.126}$ | $.372_{\pm.059}$ | $2.932_{\pm.875}$ |
| BCE | $.742_{\pm.004}$ | $.835_{\pm.003}$ | $1.501_{\pm.175}$ | $.399_{\pm.088}$ | $2.355_{\pm.434}$ | $.746_{\pm.012}$ | $.838_{\pm.010}$ | $1.443_{\pm.106}$ | $.427_{\pm.033}$ | $2.169_{\pm.264}$ |
| Dice | $.779_{\pm.018}$ | $.863_{\pm.014}$ | $1.293_{\pm.116}$ | $.471_{\pm.045}$ | $1.696_{\pm.275}$ | $.776_{\pm.015}$ | $.861_{\pm.012}$ | $1.335_{\pm.136}$ | $.445_{\pm.079}$ | $2.146_{\pm.832}$ |
| Focal | $.737_{\pm.008}$ | $.832_{\pm.006}$ | $1.404_{\pm.168}$ | $.471_{\pm.113}$ | $2.147_{\pm.331}$ | $.745_{\pm.011}$ | $.837_{\pm.009}$ | $1.410_{\pm.139}$ | $.450_{\pm.075}$ | $2.141_{\pm.400}$ |
| Jaccard | $.761_{\pm.013}$ | $.850_{\pm.011}$ | $1.289_{\pm.181}$ | $.502_{\pm.088}$ | $1.988_{\pm.641}$ | $.766_{\pm.020}$ | $.854_{\pm.015}$ | $1.271_{\pm.145}$ | $.521_{\pm.051}$ | $1.623_{\pm.313}$ |
| Tversky | $.752_{\pm.020}$ | $.843_{\pm.016}$ | $1.309_{\pm.086}$ | $.517_{\pm.052}$ | $1.638_{\pm.369}$ | $.768_{\pm.014}$ | $.856_{\pm.011}$ | $1.265_{\pm.129}$ | $.521_{\pm.048}$ | $1.702_{\pm.298}$ |
| WCE | $.741_{\pm.013}$ | $.835_{\pm.011}$ | $1.417_{\pm.069}$ | $.462_{\pm.019}$ | $2.100_{\pm.498}$ | $.733_{\pm.009}$ | $.828_{\pm.007}$ | $1.441_{\pm.031}$ | $.444_{\pm.039}$ | $2.326_{\pm.848}$ |
| ABW | — | — | — | — | — | — | — | — | — | — |
| Skea-topo | — | — | — | — | — | $.599_{\pm.121}$ | $.674_{\pm.169}$ | $1.602_{\pm.113}$ | $.265_{\pm.250}$ | $5.791_{\pm4.492}$ |
| RMI | $.787_{\pm.012}$ | $.869_{\pm.010}$ | $1.282_{\pm.127}$ | $.449_{\pm.064}$ | $1.797_{\pm.489}$ | $.785_{\pm.009}$ | $.867_{\pm.007}$ | $1.279_{\pm.096}$ | $.448_{\pm.047}$ | $1.884_{\pm.340}$ |
| clDice | $.482_{\pm.094}$ | $.598_{\pm.085}$ | $2.010_{\pm.119}$ | $.207_{\pm.129}$ | $9.813_{\pm8.153}$ | $.509_{\pm.063}$ | $.617_{\pm.066}$ | $1.912_{\pm.157}$ | $.238_{\pm.161}$ | $15.704_{\pm11.080}$ |
| Sensitive | $.763_{\pm.008}$ | $.850_{\pm.007}$ | $1.440_{\pm.060}$ | $.376_{\pm.039}$ | $2.824_{\pm.562}$ | $.757_{\pm.018}$ | $.846_{\pm.015}$ | $1.444_{\pm.040}$ | $.380_{\pm.028}$ | $3.134_{\pm.782}$ |
| CWMI-Real | $.788_{\pm.016}$ | $.870_{\pm.012}$ | $1.104_{\pm.175}$* | $.582_{\pm.075}$ | $1.309_{\pm.299}$* | $.787_{\pm.026}$ | $.870_{\pm.019}$ | $1.072_{\pm.113}$** | $.594_{\pm.057}$* | $1.398_{\pm.468}$ |
| CWMI | $\mathbf{.798_{\pm.012}}$* | $\mathbf{.878_{\pm.009}}$* | $\mathbf{1.032_{\pm.176}}$* | $\mathbf{.613_{\pm.082}}$* | $\mathbf{1.079_{\pm.382}}$* | $\mathbf{.795_{\pm.015}}$* | $\mathbf{.875_{\pm.012}}$* | $\mathbf{1.007_{\pm.162}}$** | $\mathbf{.622_{\pm.062}}$** | $\mathbf{1.219_{\pm.410}}$* |

**MASS ROAD**

| Methods | UNet | | | | | AttenUNet | | | | |
|---|---|---|---|---|---|---|---|---|---|---|
| | mIoU↑ | mDice↑ | VI↓ | ARI↑ | HD↓ | mIoU↑ | mDice↑ | VI↓ | ARI↑ | HD↓ |
| CE | $.733_{\pm.015}$ | $.826_{\pm.013}$ | $3.473_{\pm.608}$ | $.194_{\pm.083}$ | $1.927_{\pm3.301}$ | $.640_{\pm.209}$ | $.725_{\pm.195}$ | $.906_{\pm.107}$ | $.406_{\pm.327}$ | $7.914_{\pm6.637}$ |
| BCE | $.689_{\pm.011}$ | $.794_{\pm.009}$ | $1.708_{\pm.265}$ | $.560_{\pm.059}$ | $12.792_{\pm3.684}$ | $.619_{\pm.222}$ | $.708_{\pm.208}$ | $.986_{\pm.139}$ | $.365_{\pm.358}$ | $4.354_{\pm1.869}$ |
| Dice | $.761_{\pm.011}$ | $.849_{\pm.009}$ | $1.601_{\pm.138}$ | $.558_{\pm.042}$ | $11.189_{\pm4.532}$ | $.636_{\pm.197}$ | $.723_{\pm.187}$ | $.954_{\pm.056}$ | $.396_{\pm.322}$ | $7.335_{\pm4.843}$ |
| Focal | $.671_{\pm.015}$ | $.780_{\pm.013}$ | $1.717_{\pm.150}$ | $.556_{\pm.042}$ | $11.576_{\pm1.950}$ | $.581_{\pm.252}$ | $.672_{\pm.240}$ | $1.043_{\pm.109}$ | $.336_{\pm.369}$ | $5.040_{\pm2.802}$ |
| Jaccard | $.759_{\pm.008}$ | $.848_{\pm.006}$ | $1.332_{\pm.041}$ | $.630_{\pm.008}$ | $11.274_{\pm2.805}$ | $.638_{\pm.202}$ | $.725_{\pm.190}$ | $.888_{\pm.076}$ | $.406_{\pm.331}$ | $6.758_{\pm4.431}$ |
| Tversky | $.755_{\pm.009}$ | $.845_{\pm.007}$ | $1.379_{\pm.196}$ | $.618_{\pm.061}$ | $11.501_{\pm4.470}$ | $.627_{\pm.195}$ | $.717_{\pm.186}$ | $.924_{\pm.102}$ | $.384_{\pm.314}$ | $7.521_{\pm4.493}$ |
| WCE | $.650_{\pm.016}$ | $.762_{\pm.014}$ | $1.726_{\pm.103}$ | $.556_{\pm.004}$ | $1.845_{\pm.470}$ | $.823_{\pm.016}$ | $.898_{\pm.011}$ | $.873_{\pm.097}$ | $.705_{\pm.023}$ | $2.590_{\pm.602}$ |
| ABW | $.685_{\pm.014}$ | $.791_{\pm.012}$ | $1.678_{\pm.124}$ | $.566_{\pm.030}$ | $11.654_{\pm4.661}$ | $.762_{\pm.019}$ | $.858_{\pm.013}$ | $1.472_{\pm.103}$ | $.614_{\pm.037}$ | $3.623_{\pm.887}$ |
| Skea-topo | $.620_{\pm.095}$ | $.734_{\pm.085}$ | $2.236_{\pm.824}$ | $.463_{\pm.159}$ | $11.653_{\pm5.620}$ | $.784_{\pm.016}$ | $.873_{\pm.010}$ | $1.299_{\pm.086}$ | $.645_{\pm.022}$ | $3.363_{\pm.609}$ |
| RMI | $.766_{\pm.008}$ | $.852_{\pm.006}$ | $1.824_{\pm.263}$ | $.499_{\pm.065}$ | $11.632_{\pm3.918}$ | $.839_{\pm.013}$ | $.908_{\pm.008}$ | $.819_{\pm.071}$ | $.727_{\pm.032}$ | $2.587_{\pm.435}$ |
| clDice | $.645_{\pm.051}$ | $.755_{\pm.045}$ | $2.162_{\pm.467}$ | $.476_{\pm.081}$ | $15.461_{\pm4.999}$ | $.800_{\pm.013}$ | $.884_{\pm.009}$ | $1.004_{\pm.113}$ | $.684_{\pm.026}$ | $3.146_{\pm.286}$ |
| Sensitive | $.734_{\pm.022}$ | $.827_{\pm.018}$ | $2.933_{\pm.982}$ | $.305_{\pm.159}$ | $11.292_{\pm3.828}$ | $.824_{\pm.018}$ | $.899_{\pm.011}$ | $.880_{\pm.108}$ | $.702_{\pm.041}$ | $2.757_{\pm.398}$ |
| CWMI-Real | $.765_{\pm.010}$ | $.853_{\pm.008}$ | $1.231_{\pm.183}$ | $.650_{\pm.050}$ | $1.836_{\pm4.994}$ | $.846_{\pm.012}$ | $.913_{\pm.008}$ | $.766_{\pm.067}$** | $.736_{\pm.021}$* | $2.504_{\pm.528}$ |
| CWMI | $\mathbf{.767_{\pm.010}}$ | $\mathbf{.854_{\pm.008}}$ | $\mathbf{1.148_{\pm.166}}$* | $\mathbf{.667_{\pm.047}}$* | $\mathbf{1.326_{\pm4.109}}$ | $.839_{\pm.015}$ | $.908_{\pm.010}$ | $\mathbf{.762_{\pm.084}}$*** | $\mathbf{.728_{\pm.035}}$** | $\mathbf{2.747_{\pm.680}}$ |

*Table 1.* Quantitative results of different loss functions across the four datasets and two neural network models. The **bold** numbers indicate the best performance for each metric, while the underlined numbers denote the second-best performance. A hyphen ("-") indicates cases where the model did not converge. For the loss metrics CWMI and CWMI-real, the statistical significance is marked with asterisks: $*p < .05$, $**p < .01$, and $***p < .001$, where p is the maximum p-value of the student's t tests against all the baseline loss functions.

size of 10. Early stopping based on mIoU was employed to select the best model. Training was conducted on an NVIDIA A100 GPU using the Google Colab runtime.

**Data Augmentation and Evaluation Metrics** Random flips and rotations were applied to all datasets to improve generalization. For SNEMI3D and MASS ROAD, images were randomly cropped to $512 \times 512$, while for GlaS, images were cropped to $448 \times 576$ to standardize input sizes. No cropping was performed for DRIVE due to its uniform image dimensions.

Performance was evaluated using five metrics: mIoU and mDice for regional precision, variation of information (VI) (Nunez-Iglesias et al., 2013) and adjusted Rand index (ARI) (Vinh et al., 2009) for clustering precision, and Hausdorff distance (HD) for boundary and topological accuracy. These metrics provide a comprehensive assessment of both regional overlap and structural fidelity.

| | **SNEMI3D VMUNet** | | | | |
|---|---|---|---|---|---|
| Methods | mIoU↑ | mDice↑ | VI↓ | ARI↑ | HD↓ |
| CE | $.723_{\pm.026}$ | $.829_{\pm.020}$ | $2.577_{\pm1.211}$ | $.414_{\pm.173}$ | $1.709_{\pm.235}$ |
| BCE | $.739_{\pm.004}$ | $.843_{\pm.002}$ | $1.499_{\pm.057}$ | $.594_{\pm.010}$ | $1.274_{\pm.231}$ |
| Dice | $.725_{\pm.029}$ | $.831_{\pm.021}$ | $2.735_{\pm1.213}$ | $.402_{\pm.165}$ | $1.767_{\pm.720}$ |
| Focal | $.725_{\pm.010}$ | $.833_{\pm.007}$ | $1.677_{\pm.141}$ | $.563_{\pm.025}$ | $1.699_{\pm.372}$ |
| Jaccard | $.725_{\pm.019}$ | $.831_{\pm.014}$ | $1.953_{\pm.541}$ | $.499_{\pm.075}$ | $1.716_{\pm.566}$ |
| Tversky | $.737_{\pm.016}$ | $.840_{\pm.012}$ | $1.619_{\pm.324}$ | $.555_{\pm.045}$ | $1.560_{\pm.408}$ |
| WCE | $.714_{\pm.005}$ | $.826_{\pm.004}$ | $1.743_{\pm.026}$ | $.552_{\pm.006}$ | $1.432_{\pm.169}$ |
| ABW | $.678_{\pm.003}$ | $.800_{\pm.003}$ | $2.076_{\pm.030}$ | $.490_{\pm.005}$ | $1.572_{\pm.021}$ |
| RMI | $\underline{.762}_{\pm.010}$ | $\underline{.857}_{\pm.007}$ | $\underline{1.419}_{\pm.183}$ | $\underline{.588}_{\pm.030}$ | $\underline{1.071}_{\pm.306}$ |
| clDice | $.714_{\pm.002}$ | $.824_{\pm.001}$ | $1.754_{\pm.062}$ | $.542_{\pm.003}$ | $1.990_{\pm.262}$ |
| sensitive | — | — | — | — | — |
| CWMI | $\mathbf{.783}_{\pm.004}^{*}$ | $\mathbf{.872}_{\pm.003}^{*}$ | $\mathbf{.982}_{\pm.106}^{*}$ | $\mathbf{.660}_{\pm.022}^{*}$ | $\mathbf{.629}_{\pm.073}$ |

*Table 2.* Quantitative results of different loss functions on SNEMI3D dataset and VMUNet model. The **bold** numbers indicate the best performance for each metric, while the underlined numbers denote the second-best performance. A hyphen ("-") indicates cases where the model did not converge. For the loss metrics CWMI, the statistical significance is marked with asterisks: $*p < .05$, where p is the maximum p-value of the student's t tests against all the baseline loss functions.

### 4.2. Quantitative and qualitative results

As shown in Table 1, the proposed CWMI loss outperforms other loss functions across the majority of evaluation metrics for all datasets using both U-Net and Attention U-Net architectures. Statistically, CWMI achieves significantly superior performance on all metrics for the SNEMI3D and DRIVE datasets, as well as on clustering-based metrics (VI and ARI) for the GlaS and MASS ROAD datasets. Moreover, CWMI consistently outperforms its real-valued variant, underscoring the importance of incorporating phase information within the steerable pyramid decomposition. For the Mamba-based architecture, VMUNet, CWMI also significantly outperforms baseline loss functions on the SNEMI3D dataset (Table 2). Qualitative results from SNEMI3D (Figure 3), GlaS (Figure 4), DRIVE (Figure 5), and MASS ROAD (Figure 6) further illustrate CWMI's effectiveness

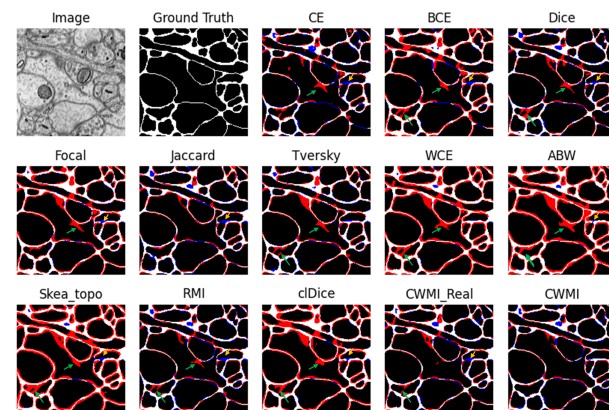

*Figure 3.* Qualitative results of different loss functions on the SNEMI3D dataset. Red: false positive regions; Blue: false negative regions. Green arrow: challenging false positive and Orange arrow: challenging false negative that are successfully addressed by CWMI.

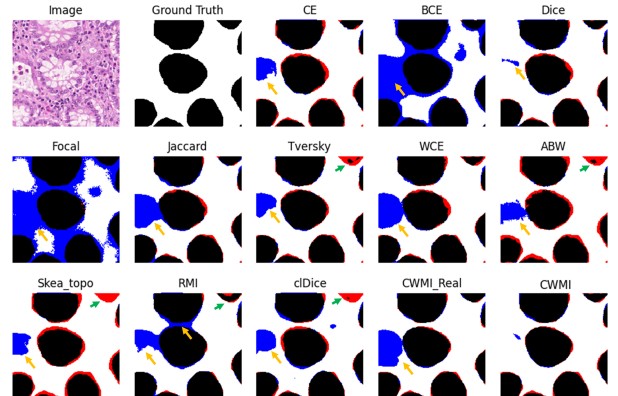

*Figure 4.* Qualitative results of different loss functions on the GlaS dataset.

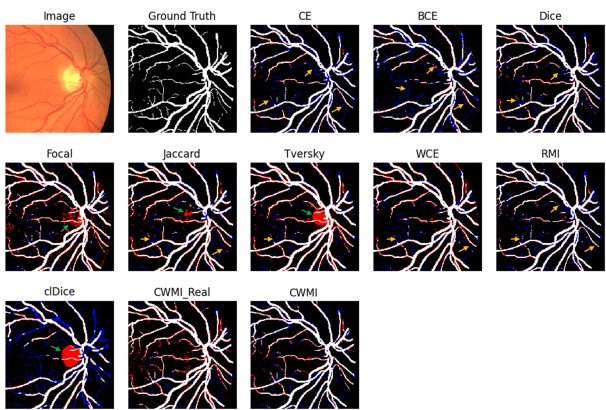

*Figure 5.* Qualitative results of different loss functions on the DRIVE dataset.

in mitigating challenging false positive and false negative segmentation errors that persist with other state-of-the-art loss functions.

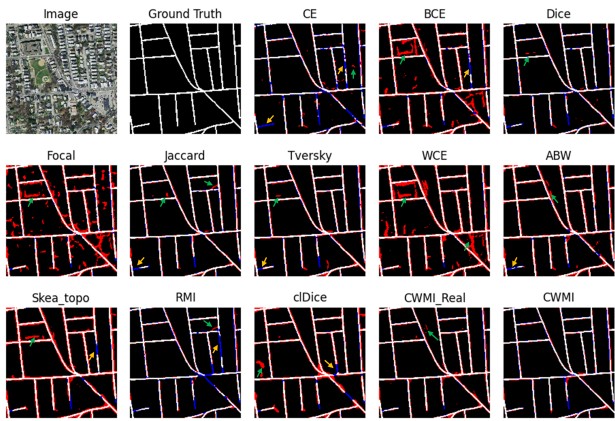

*Figure 6.* Qualitative results of different loss functions on the MASS ROAD dataset.

## 4.3. Ablation studies

In this section, all ablation experiments were performed on the SNEMI3D dataset with U-Net model.

**Mutual Information (MI) vs $L_1$, $L_2$ Distance and Structural Similarity (SSIM)** As previously discussed, various metrics based on wavelet transforms have been developed for evaluating image similarity and guiding loss functions. To assess the advantages of mutual information (MI) over commonly used metrics such as $L_1$, $L_2$, and SSIM, we conducted a comparative analysis. The experimental results, summarized in Table 3, demonstrate that MI consistently outperforms $L_1$, $L_2$ distance, and SSIM across multiple evaluation metrics. Unlike $L_1$ and $L_2$, which focus on pixel-wise intensity differences, and SSIM, which emphasizes structural similarity, MI captures joint statistical dependencies between features in each direction. This ability provides a more robust representation of structural differences between predictions and labels, particularly in complex segmentation tasks.

| Methods | mIoU↑ | mDice↑ | VI↓ | ARI↑ | HD↓ |
|---|---|---|---|---|---|
| L1 | $.773_{\pm.002}$ | $.866_{\pm.001}$ | $1.22_{\pm.05}$ | $.632_{\pm.007}$ | $.70_{\pm.08}$ |
| L2 | $.774_{\pm.003}$ | $.867_{\pm.002}$ | $1.23_{\pm.06}$ | $.633_{\pm.011}$ | $.76_{\pm.13}$ |
| SSIM | $.468_{\pm.032}$ | $.613_{\pm.020}$ | $5.96_{\pm.96}$ | $.081_{\pm.030}$ | $9.31_{\pm1.64}$ |
| CWMI | $\mathbf{.779}_{\pm.004}$ | $\mathbf{.870}_{\pm.003}$ | $\mathbf{1.16}_{\pm.09}$ | $\mathbf{.640}_{\pm.005}$ | $\mathbf{.69}_{\pm.04}$ |

*Table 3.* Quantitative results comparing mutual information (MI), $L_1$, $L_2$ distance, and structural similarity (SSIM) based on U-Net with SNEMI3D. The **bold** numbers indicate the best performance for each metric.

**Contribution of different components of complex steerable pyramid** To better understand the role of individual components in the CWMI loss, we conducted an ablation experiment evaluating the performance of variants based on the mutual information loss of magnitude-only, phase-only, and real-only representations, as shown in Table 4. The CWMI-Phase variant consistently underperformed across all metrics, suggesting that phase information alone

is insufficient for reliable segmentation. The CWMI-Mag variant, by contrast, achieved the best performance on clustering metrics (VI and ARI), indicating its strong capacity to capture regional structures. The full CWMI, combining both magnitude and phase via the complex representation, yielded the best results on pixel-wise accuracy metrics (mIoU and mDice) and topological metric (HD), while ranking second on VI and ARI, demonstrating a well-rounded performance across different dimensions of segmentation quality. The CWMI-Real variant performed worse than both CWMI and CWMI-Mag on most metrics, although it outperformed CWMI-Phase, further supporting the necessity of incorporating both magnitude and phase components. These results highlight the critical role of magnitude in structural segmentation, while also demonstrating that the full complex formulation of CWMI achieves superior generalization across diverse evaluation criteria.

| Methods | mIoU↑ | mDice↑ | VI↓ | ARI↑ | HD↓ |
|---|---|---|---|---|---|
| CWMI-Real | $.776_{\pm.004}$ | $.868_{\pm.003}$ | $1.211_{\pm.048}$ | $.635_{\pm.011}$ | $.774_{\pm.065}$ |
| CWMI-Mag | $.775_{\pm.004}$ | $.867_{\pm.003}$ | $\mathbf{.908}_{\pm.051}$ | $\mathbf{.660}_{\pm.009}$ | $.735_{\pm.065}$ |
| CWMI-Phase | $.694_{\pm.008}$ | $.810_{\pm.006}$ | $2.487_{\pm.534}$ | $.458_{\pm.061}$ | $2.000_{\pm.258}$ |
| CWMI | $\mathbf{.778}_{\pm.004}$ | $\mathbf{.869}_{\pm.003}$ | $\underline{1.164}_{\pm.079}$ | $\underline{.638}_{\pm.016}$ | $\mathbf{.720}_{\pm.066}$ |

*Table 4.* Quantitative comparison of mutual information loss across different components of the complex steerable pyramid decomposition. CWMI-Real: real component; CWMI-Mag: magnitude component; CWMI-Phase: phase component; CWMI: full complex representation (combining real and imaginary components). Results are based on a U-Net model trained on the SNEMI3D dataset. **Bold** values indicate the best performance, and underlined values denote the second-best for each evaluation metric.

**Impact of Regularization Parameter $\lambda$** To evaluate the sensitivity of CWMI to the regularization weight $\lambda$, we experimented with $\lambda = 0.1, 0.5, 0.9$. As shown in Table 5, performance across all evaluation metrics remained largely stable, with only marginal variations. This robustness suggests that CWMI provides complementary information to the cross-entropy term and is not overly sensitive to the regularization strength. Additionally, we found the inclusion of the cross-entropy term to be crucial for directional learning, as mutual information is inherently symmetric and cannot distinguish between correct and inverted label assignments.

**Impact of Decomposition Level $N$ and Number of Orientations $K$** Theoretically, the steerable pyramid can decompose an image into very high levels with an infinite number of orientations, provided the input image is sufficiently large. However, does deeper decomposition or a higher number of orientations improve feature extraction and loss computation? To address this, we analyzed the impact of $N$ (decomposition level) and $K$ (number of orientations) on the performance of CWMI, as shown in Table 3. Interestingly, both $N$ and $K$ achieved optimal performance at relatively low values, suggesting that the critical information for segmentation is concentrated in relatively high-frequency regions. This observation supports the idea

that emphasizing high-frequency subbands refines small instances and narrow boundaries, while lower-frequency components from deeper decompositions contribute less informative features compared to higher-frequency ones.

Knowing that the first four layers of decomposition are crucial, we further performed a layer ablation experiment (Table 3, Layer Ablation), where only selected layers were used to compute mutual information while discarding the remaining layers. Our results show that the third layer outperforms other layers in region-based (mIoU, mDice) and cluster-based metrics (VI, ARI), while the fourth layer achieves the best performance in topological metrics (HD). These findings suggest that features extracted from multiple layers are essential for achieving high performance across all evaluation metrics, highlighting the importance of incorporating information from both mid- and high-frequency subbands in the segmentation task.

| λ ablation | | | | |
|---|---|---|---|---|
| | mIoU↑ | mDice↑ | VI↓ | ARI↑ | HD↓ |
| λ=0.1 | **.778**$_{\pm.004}$ | **.869**$_{\pm.003}$ | 1.188$_{\pm.075}$ | .636$_{\pm.008}$ | **.703**$_{\pm.034}$ |
| λ=0.5 | .778$_{\pm.006}$ | .869$_{\pm.004}$ | 1.176$_{\pm.076}$ | **.639**$_{\pm.014}$ | .729$_{\pm.078}$ |
| λ=0.9 | .777$_{\pm.003}$ | .869$_{\pm.002}$ | 1.219$_{\pm.073}$ | .636$_{\pm.013}$ | .740$_{\pm.061}$ |

| K ablation | | | | |
|---|---|---|---|---|
| | mIoU↑ | mDice↑ | VI↓ | ARI↑ | HD↓ |
| K=2 | .777$_{\pm.006}$ | .868$_{\pm.004}$ | 1.18$_{\pm.10}$ | .637$_{\pm.028}$ | .79$_{\pm.02}$ |
| K=4 | **.779**$_{\pm.004}$ | **.870**$_{\pm.003}$ | **1.16**$_{\pm.09}$ | **.640**$_{\pm.005}$ | **.69**$_{\pm.04}$ |
| K=8 | .776$_{\pm.003}$ | .868$_{\pm.002}$ | 1.20$_{\pm.06}$ | .635$_{\pm.009}$ | .76$_{\pm.08}$ |
| K=12 | .772$_{\pm.005}$ | .865$_{\pm.003}$ | 1.33$_{\pm.05}$ | .621$_{\pm.004}$ | .81$_{\pm.08}$ |

| N ablation | | | | |
|---|---|---|---|---|
| | mIoU↑ | mDice↑ | VI↓ | ARI↑ | HD↓ |
| N=2 | .760$_{\pm.002}$ | .857$_{\pm.001}$ | 1.30$_{\pm.04}$ | .618$_{\pm.009}$ | 1.20$_{\pm.12}$ |
| N=4 | **.777**$_{\pm.002}$ | **.869**$_{\pm.001}$ | **1.21**$_{\pm.10}$ | **.634**$_{\pm.017}$ | **.79**$_{\pm.190}$ |
| N=6 | .768$_{\pm.003}$ | .862$_{\pm.002}$ | 1.29$_{\pm.05}$ | .618$_{\pm.004}$ | .88$_{\pm.18}$ |

| Layer ablation | | | | |
|---|---|---|---|---|
| | mIoU↑ | mDice↑ | VI↓ | ARI↑ | HD↓ |
| 1st layer | .738$_{\pm.008}$ | .843$_{\pm.006}$ | 1.61$_{\pm.07}$ | .576$_{\pm.015}$ | 1.38$_{\pm.14}$ |
| 2nd layer | .762$_{\pm.008}$ | .858$_{\pm.006}$ | 1.25$_{\pm.08}$ | .624$_{\pm.016}$ | 1.12$_{\pm.19}$ |
| 3rd layer | **.774**$_{\pm.003}$ | **.867**$_{\pm.002}$ | **1.23**$_{\pm.10}$ | **.638**$_{\pm.020}$ | .82$_{\pm.12}$ |
| 4th layer | .766$_{\pm.004}$ | .861$_{\pm.003}$ | 1.39$_{\pm.09}$ | .614$_{\pm.012}$ | **.76**$_{\pm.04}$ |

*Table 5.* The impact of regularization parameter λ, decomposition level N, and orientation number K on CWMI performance on U-Net with SNEMI3D. In the layer ablation test, only the selected layer is computed for mutual information and all other layers are discard. The **bold** numbers indicate the best performance for each metric.

**Computational complexity analysis** For an input image of size $H \times W$ with $K$ orientation decompositions, the computational complexity of the CWMI loss function is analyzed as follows.

First, the forward Fourier transform has a time complexity of $O(HW \log(HW))$. For the first layer of decomposition, the operations include: Band-pass filtering: $O(HWK)$; Inverse Fourier transform: $O(HW \log(HW)K)$; and mutual information computation: $O(HWK^2)$. Summing these terms, the total complexity for the first decomposition layer is: $O(HW(K \log(HW) + K^2))$ For subsequent decomposition layers, the image size reduces by a factor of four at each step, meaning the second layer pro-

cesses an image of size $HW/4$, and the third layer processes $HW/16$, and so on. Thus, the total computational complexity follows a geometric series with a superior bound $O\left(\frac{4}{3}HW(K \log(HW) + K^2)\right)$

From our ablation experiments, the optimal number of orientations is $K = 4$, which is relatively small. Hence the complexity of CWMI is linear to $O(HW \log(HW))$, which remains scalable for high-resolution images. Compared to topology-aware losses such as RMI, clDice, or Hausdorff Distance Loss, CWMI achieves better structural and boundary preservation with lower computational overhead, as shown in Table 4.

| | Epoch Time (s) | $\Delta t$ to CE (s) |
|---|---|---|
| CE | 2.04 | .00 |
| BCE | 2.02 | -0.02 |
| Dice | 2.01 | -0.03 |
| Focal | 2.01 | -0.03 |
| Jaccard | 2.02 | -0.02 |
| Tversky | 2.01 | -0.03 |
| WCE | 2.15 | 0.12 |
| ABW | 2.33 | 0.30 |
| Skea-topo | 2.72 | 0.69 |
| RMI | 2.28 | 0.25 |
| clDice | 2.37 | 0.34 |
| CWMI-Real | 2.18 | 0.15 |
| CWMI | 2.27 | 0.23 |

*Table 6.* Training time per epoch and relative change to CE baseline for various loss functions on U-Net model with SNEMI3D dataset.

## 5. Conclusion

In this study, we introduced Complex Wavelet Mutual Information (CWMI) loss, a novel loss function for semantic segmentation that leverages the multiscale, multi-orientation decomposition capabilities of the complex steerable pyramid. By integrating mutual information across wavelet subbands, CWMI effectively captures high-dimensional dependencies and local structural features, including critical phase information, which are essential for accurate segmentation. Extensive experiments on four challenging datasets demonstrate that CWMI consistently outperforms state-of-the-art loss functions across most evaluation metrics, particularly in segmenting small instances and narrow boundaries, while introducing minimal computational overhead. These results highlight CWMI as a robust and versatile loss function that effectively addresses key challenges in segmentation, such as class and instance imbalance, boundary precision, and topological consistency.

Beyond semantic segmentation, the core principles of CWMI—multiscale feature extraction and structural consistency—suggest its potential applicability to a broader range of computer vision and machine learning tasks, such as image-to-image translation and super-resolution, which we leave for future exploration.

Although extensively studied on 2D images, extending it to multi-class and 3D segmentation is theoretically feasible but necessitates further validation in future research.

## Acknowledgements

This study was conducted without external funding. I would like to thank the reviewers for their constructive feedback, which significantly improved the quality of this work. I am especially grateful to my wife, Di Duan, for her unwavering support of my passion for machine learning. I thank Xinzi He for valuable discussions that contributed to the revision of this study.

## Impact Statement

The proposed Complex Wavelet Mutual Information (CWMI) loss introduces a novel approach to structural-aware learning in deep neural networks. By leveraging multiscale decomposition through the complex steerable pyramid and mutual information across frequency subbands, CWMI enables improved segmentation performance, particularly for small-scale structures and thin boundaries. Our empirical results demonstrate significant improvements in both pixel-wise and topological accuracy across multiple datasets. Beyond segmentation, CWMI has the potential to generalize to a wide range of real-world applications, including medical imaging, autonomous driving, satellite-based environmental monitoring, and industrial defect detection.

Overall, CWMI provides a **computationally efficient**, **structure-aware** loss function that enhances segmentation performance while maintaining practical scalability. By extending its application beyond segmentation, we aim to contribute to the broader field of generative modeling, object detection, and self-supervised learning in deep neural networks.

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
