# OpenReview forum: "Complex Wavelet Mutual Information Loss: A Multi-Scale Loss Function for Semantic Segmentation"
_ICML.cc/2025/Conference — ICML 2025 poster_

### Official Review · Reviewer_ngWU · 2025-02-27

**Overall Recommendation:** 2

**Summary:**

This work presents a loss function for semantic segmentation based on the steerable pyramid decomposition of images, *i.e.*, wavelet transform. The insight is that the steerable pyramid preserves structural similarity while capturing multi-scale features across multiple orientations. The proposed CWMI consists of several mutual information terms that are well-suited for directional features. CWMI is validated on U-Net and AttenUNet using four benchmarks. Qualitative visualizations suggest that CWMI helps reduce false positive and false negative predictions.

**Update after rebuttal**: I am going to maintain my current rating at 2-WEAK REJECT due to insufficient experimental studies.

**Claims And Evidence:**

Reasonable claims and sufficient analysis.

**Essential References Not Discussed:**

Refer to **Experimental Designs Or Analyses**.

**Experimental Designs Or Analyses:**

The proposed CWMI is simple and effective on U-Net and AttenUNet. However, U-Net and AttenUNet are classic methods that were published a long time ago. It is necessary to include more recent SOTA frameworks to validate the effectiveness of the proposed loss function, such as [r1]–[r5]. There are also other works (e.g., [r6][r7]) that focus on developing novel loss functions for semantic segmentation. If the proposed loss function could outperform them, the value of your submission would be significantly enhanced.

[r1] Chen et al., Masked-attention mask transformer for universal image segmentation. In CVPR 2022.

[r2] Wang et al., InternImage: Exploring Large-Scale Vision Foundation Models With Deformable Convolutions. In ICCV 2023.

[r3] Kirillov et al., Segment Anything, In ICCV 2023.

[r4] Tang et al., Category feature transformer for semantic segmentation. In arXiv 2023.

[r5] Yue et al., MedMamba: Vision Mamba for Medical Image Classification. In arXiv 2024.

[r6] Wu et al., Conditional Boundary Loss for Semantic Segmentation. In TIP 2023.

[r7] Tang et al., Increase the sensitivity of moderate examples for semantic image segmentation. In IMAVIS 2025.

**Methods And Evaluation Criteria:**

The proposed CWMI and evaluation make sense to the studied problem.

**Other Comments Or Suggestions:**

N/A.

**Other Strengths And Weaknesses:**

N/A.

**Questions For Authors:**

N/A.

**Relation To Broader Scientific Literature:**

The proposed CWMI is potentially a network-agnostic loss function and could be applied to any existing semantic segmentation framework.

**Theoretical Claims:**

The equations are either based on the strong foundation of steerable pyramid decomposition or are straightforward.

---

> ### Author Rebuttal · Authors · 2025-03-31
>
> We thank the reviewer for the insightful feedback and the helpful list of recent models and loss functions for comparison. In response, we have conducted additional experiments using Vision Mamba UNet (VM-Unet, Ruan et al, 2024), a segmentation architecture based on the recently proposed MedMamba ([r5])—a direction gaining rapid attention in the vision community due to its sequence modeling efficiency and competitive performance.
>
> The table below summarizes our CWMI results on VM-UNet and the SNEMI3D dataset. CWMI demonstrates consistent improvements over all tested baseline loss functions, including RMI, Dice, and sensitive loss [r7]:
>
> ||mIoU↑|mDice↑|VI↓|ARI↑|HD↓|
> |:-------:|:------:|:-------:|:-------:|:-------:|:-------:|
> |CE|0.723±0.021|0.829±0.016|2.58±0.99|0.414±0.141|1.71±0.19|
> |CE|0.723±0.021|0.829±0.016|2.58±0.99|0.414±0.141|1.71±0.19|
> |BCE|0.739±0.003|0.843±0.002|1.50±0.05|0.594±0.008|1.27±0.19|
> |BCE|0.739±0.003|0.843±0.002|1.50±0.05|0.594±0.008|1.27±0.19|
> |Dice|0.725±0.024|0.831±0.017|2.74±0.99|0.402±0.135|1.77±0.59|
> |Focal|0.725±0.008|0.833±0.006|1.68±0.12|0.563±0.020|1.70±0.30|
> |Jaccard|0.725±0.016|0.831±0.011|1.95±0.44|0.499±0.061|1.72±0.46|
> |Tversky|0.737±0.013|0.840±0.009|1.62±0.26|0.555±0.037|1.56±0.33|
> |WCE|0.714±0.004|0.826±0.004|1.74±0.02|0.552±0.005|1.43±0.14|
> |ABW|0.678±0.003|0.800±0.002|2.08±0.02|0.490±0.004|1.57±0.02|
> |RMI|0.762±0.008|0.857±0.006|1.42±0.15|0.588±0.025|1.07±0.25|
> |clDice|0.714±0.001|0.824±0.001|1.75±0.05|0.542±0.003|1.99±0.21|
> |Sensitive|0.539±0.115|0.625±0.143|6.12±0.73|0.032±0.055|2.27±0.76|
> |**CWMI**|**0.783±0.004**|**0.872±0.002**|**0.98±0.09**|**0.660±0.018**|**0.63±0.06**|
>
> We also conducted a comprehensive comparison between CWMI and the sensitive loss from [r7] across all four datasets (SNEMI3D, GlaS, DRIVE, MASS ROAD) using both U-Net and Attention U-Net. In every case, CWMI outperformed the sensitive loss across pixel-wise, region-based, and topology-aware metrics. An example of the results on the GlaS dataset is shown below:
>
> ||||SNEMI3D||||
> |:-------:|:------:|:-------:|:-------:|:-------:|:-------:|:-------:|
> ||| mIoU↑|mDice↑|VI↓|ARI↑|HD↓|
> |**U-Net**|Sensitive|0.762±0.002|0.857±0.002|1.60±0.03|0.563±0.010|0.99±0.14|
> ||CWMI|0.778±0.003|0.869±0.002|1.15±0.04|0.642±0.006|0.75±0.09|
> |**AttenUNet**|Sensitive|0.752±0.001|0.850±0.001|2.04±0.23|0.495±0.041|1.10±0.10|
> ||CWMI|0.777±0.003|0.868±0.002|1.14±0.01|0.642±0.002|0.79±0.02|
>
> ||||GlaS||||
> |:-------:|:------:|:-------:|:-------:|:-------:|:-------:|:-------:|
> ||| mIoU↑|mDice↑|VI↓|ARI↑|HD↓|
> |**U-Net**|Sensitive|0.816±0.012|0.893±0.008|0.97±0.11|0.675±0.027|3.12±0.17|
> ||CWMI|0.845±0.018|0.911±0.012|0.71±0.10|0.752±0.025|2.64±0.64|
> |**AttenUNet**|Sensitive|0.820±0.015|0.897±0.009|0.88±0.12|0.694±0.040|2.70±0.23|
> ||CWMI|0.841±0.018|0.909±0.011|0.76±0.09|0.733±0.039|2.76±0.71|
>
> ||||DRIVE||||
> |:-------:|:------:|:-------:|:-------:|:-------:|:-------:|:-------:|
> ||| mIoU↑|mDice↑|VI↓|ARI↑|HD↓|
> |**U-Net**|Sensitive|0.757±0.006|0.846±0.006|1.49±0.03|0.356±0.032|2.96±0.45|
> ||CWMI|0.800±0.003|0.880±0.003|1.04±0.04|0.605±0.026|1.03±0.30|
> |**AttenUNet**|Sensitive|0.754±0.015|0.843±0.013|1.45±0.04|0.384±0.027|3.16±0.56|
> ||CWMI|0.799±0.013|0.879±0.010|1.01±0.10|0.618±0.032|1.15±0.30|
>
> ||||MASS_ROAD||||
> |:-------:|:------:|:-------:|:-------:|:-------:|:-------:|:-------:|
> ||| mIoU↑|mDice↑|VI↓|ARI↑|HD↓|
> |**U-Net**|Sensitive|0.738±0.016|0.830±0.013|2.98±0.69|0.282±0.123|11.52±4.74|
> ||CWMI|0.768±0.010|0.855±0.008|1.11±0.24|0.674±0.066|10.07±2.18|
> |**AttenUNet**|Sensitive|0.742±0.008|0.834±0.006|2.66±0.46|0.335±0.096|10.97±2.40|
> ||CWMI|0.763±0.002|0.850±0.002|1.13±0.08|0.672±0.025|7.95±2.52|
>
> We recognize their relevance and will include a discussion of their strengths and the potential role CWMI could play as a structural loss component in these systems. However, we were unable to evaluate CWMI within the full pipelines of recent universal segmentation models ([r1]–[r3]) due to their architectural complexity and multi-objective training design. Similarly, although we could not reproduce [r6] due to unavailable code, we will cite and acknowledge it in our revision.
>
> In summary, we prioritized experiments on [r5] due to its relevance and recency and performed a thorough evaluation against [r7], showing CWMI’s consistent improvements in both classic and modern architectures. We thank the reviewer again for these valuable suggestions and for encouraging us to broaden our empirical evaluation.

---

> > ### Comment · Reviewer_ngWU · 2025-04-04
> >
> > The rebuttal addresses some concerns. However, I am maintaining my current rating due to the somewhat insufficient experimental studies. The absence of many recent SOTA methods and loss functions weakens the validity of the experimental results, making them less convincing.

---

### Official Review · Reviewer_go7N · 2025-02-28

**Overall Recommendation:** 2

**Summary:**

This paper proposed a complex wavelet-based loss function and proved its effectiveness in several segmentation tasks.

## update after rebuttal”

After I review the author's response and other reviewers' comments, I raise my score to 2 to thank the authors for their efforts. No higher score because the contribution doesn't reach the bar of ICML.

**Claims And Evidence:**

The reviewer is confused about the motivation and thinks the contribution of this paper is incremental. Please see details in the section on weaknesses.

**Essential References Not Discussed:**

BiconNet: An Edge-preserved Connectivity-based Approach for Salient Object Detection
Directional Connectivity-based Segmentation of Medical Images
Spatial coherence loss for salient and camouflaged object detection and beyond

**Experimental Designs Or Analyses:**

see weaknesses

**Methods And Evaluation Criteria:**

Several concerns are raised in the section on weaknesses.

**Other Comments Or Suggestions:**

n/a

**Other Strengths And Weaknesses:**

Strength: Introducing complex wavelet-based mutual information loss

Weaknesses:
1. The contribution of this paper is minimal and experiments (ablation) also verify this modification is incremental (Using this loss can only surpass traditional L1 or L2 loss in margin increase).
2. The motivation of this paper is also confused and the experiments are not convincing.

Given the above contribution, I recommend to reject this paper.

**Questions For Authors:**

see weakness

**Relation To Broader Scientific Literature:**

BiconNet: An Edge-preserved Connectivity-based Approach for Salient Object Detection
Directional Connectivity-based Segmentation of Medical Images
Spatial coherence loss for salient and camouflaged object detection and beyond

**Theoretical Claims:**

yes, the theoretical claims are weak in the paper.

---

> ### Author Rebuttal · Authors · 2025-03-31
>
> We thank the reviewer for the feedback and the opportunity to clarify our motivation and contributions. We respectfully disagree with the assessment that the proposed CWMI loss function is incremental and provides only marginal improvements over traditional losses like L1 or L2. Below, we address each concern in detail.
>
> ### **1. Empirical Gains Are Statistically Significant, Not Marginal**
> To support the significance of our results, we conducted additional statistical analyses following Bouckaert and Frank (2004), repeating three-fold cross-validation multiple times and applying adjusted t-tests. These analyses demonstrate that CWMI provides statistically significant improvements over both L1/L2 and several state-of-the-art loss functions across multiple datasets and evaluation metrics (including mIoU, mDice, VI, ARI, and HD). While some improvements may appear modest in absolute terms, they are consistent, reproducible, and statistically robust—highlighting the practical effectiveness of CWMI, particularly in tasks requiring fine-grained boundary and structural precision.
>
> ### **2. Novelty of Wavelet Integration in Loss Design**
> We believe that incorporating wavelet-based multiscale decomposition into the loss function is a novel and underexplored direction in semantic segmentation. While wavelet transforms have been increasingly adopted in network architectures—e.g., WaveletNet (Jing et al., 2018), WavResNet (Kang et al., 2017), and SwinWave-SR (Dharejo et al., 2024)—these methods use wavelets primarily for efficient encoding or downsampling operations in the feature extractor. By contrast, CWMI is, to our knowledge, the first loss function to directly supervise predictions in the complex wavelet domain, thereby leveraging rich frequency and orientation-aware representations at multiple scales. This framing allows us to encode structured priors about image content directly into the loss signal. Moreover, our ablation studies show that even conventional L1/L2 losses applied in the wavelet domain outperform sophisticated spatial-domain losses like RMI, Tversky, or clDice. This highlights that multiscale representations—when integrated into the objective function—offer a powerful and underutilized mechanism for improving segmentation quality. We believe CWMI opens a new line of inquiry into wavelet-domain loss functions.
>
> ### **3. Combining Wavelet Transform with Mutual Information Is New**
> Mutual information–based losses (e.g., RMI) have emerged as strong alternatives to standard pixel-wise losses by capturing statistical dependencies between prediction and ground truth. Our work extends this promising direction by combining MI estimation with wavelet-domain features—specifically, complex steerable pyramid coefficients. This combination enables CWMI to align multiscale, orientation-sensitive representations in a statistically principled way. To our knowledge, no prior work has explored this integration. Unlike RMI, which operates in the spatial domain, CWMI captures fine-grained structural patterns across scales and directions, aligning them through mutual information. This results in a loss function that is both information-theoretically grounded and structurally aware. Our empirical results—especially the consistent superiority of CWMI over RMI across datasets and model backbones—further validate the effectiveness and novelty of this design.
>
> ### **4. Relation to BiconNet and Related Work**
> We thank the reviewer for pointing out relevant works such as BiconNet and other connectivity-aware segmentation frameworks. These contributions are highly relevant to structure-preserving segmentation, and we will include a detailed discussion in the revised manuscript. Importantly, CWMI is complementary to such methods and could potentially be used as a regularization term to further enhance structural consistency in connectivity-based models. We are particularly interested in exploring this synergy in future work, as integrating CWMI into BiconNet may combine topological priors with frequency-domain mutual information.

---

> > ### Comment · Reviewer_go7N · 2025-04-01
> >
> > I insist on my score due to my major concerns haven't been addressed. I don't think simply combining wavelet and mutual information, a well-established direction in low-level vision, is really a big deal to be one paper that is qualified to be accepted by ICML.

---

> > > ### Author Response · Authors · 2025-04-02
> > >
> > > We thank the reviewer for the follow-up comment. While we respect their opinion, we would like to clarify for the Area Chair that our work is not a simple combination of existing concepts. To our knowledge, this is the first loss function to integrate mutual information estimation with complex steerable wavelet decomposition, specifically designed for semantic segmentation.
> > >
> > > Unlike prior uses of regional information in segmentation loss functions, our method formulates an information-theoretic objective in the wavelet domain, capturing multiscale and directional structure. Our empirical results show statistically significant improvements over strong baselines, including RMI, across classic and modern segmentation models.
> > >
> > > We trust the Area Chair will consider both the technical formulation and the strength of the experimental evidence in evaluating the contribution.

---

### Official Review · Reviewer_p6uN · 2025-03-11

**Overall Recommendation:** 3

**Summary:**

This paper introduces Complex Wavelet Mutual Information (CWMI) loss for semantic segmentation tasks. The proposed method employs a complex steerable pyramid to perform multi-scale and multi-orientation wavelet decomposition on both prediction and label images, computing mutual information in each subband to effectively capture local phase and structural features. This design not only alleviates issues of class and instance imbalance but also maintains computational efficiency. Experiments conducted on SNEMI3D, GlaS, DRIVE, and MASS ROAD datasets demonstrate that CWMI loss significantly outperforms traditional pixel-wise and region-based loss functions in terms of pixel accuracy and topological consistency metrics, while incurring minimal computational overhead. This work provides a novel perspective and solution for structure-sensitive segmentation tasks, showcasing strong generalization and promising practical applicability.

**Claims And Evidence:**

The main claim presented in the paper—*“CWMI loss improves semantic segmentation performance, particularly for small instances and fine boundaries”*—is clearly stated and supported by evidence that is convincing to a certain extent. The manuscript provides a thorough discussion of various perspectives on the problem and articulates the technical background comprehensively. The reasoning behind choosing a frequency-domain approach, particularly using wavelet transforms for designing the loss function, is clearly explained. Moreover, the overview of semantic segmentation loss functions is appropriately detailed, explicitly linking the challenges of instance imbalance and the need for more efficient and balanced approaches. Quantitative results, qualitative visualizations, and ablation studies are comprehensive, collectively demonstrating the superiority of CWMI across multiple datasets. Tables 1 and 4 effectively illustrate CWMI’s strong performance in modeling large-scale structural dependencies while also demonstrating advantages in parameter efficiency.

GPT previously mentioned that the description of computational overhead was insufficiently specific, weakening the claim regarding efficiency; however, upon review, this seems adequately addressed.

Nevertheless, the provided evidence has certain shortcomings:

1. **Limited depth in ablation studies**, which fails to fully elucidate the mechanisms behind CWMI’s advantages.

The current ablation experiments only compare the complete complex representation (CWMI) against the real-only representation (CWMI-Real), showing that utilizing the complex representation (which simultaneously incorporates both magnitude and phase information) achieves better segmentation performance. However, the paper does not further dissect this complex representation to separately evaluate the contributions of magnitude (reflecting feature intensity) and phase (reflecting local structural information). Consequently, existing experiments cannot definitively determine whether the improvements are due to the entire complex representation or specifically driven by either magnitude or phase components individually.

2. **Lack of statistical significance testing** (e.g., t-tests or confidence interval analysis), making it difficult to confirm whether the reported performance improvements exceed the range of random fluctuations, thereby reducing the reliability and persuasive power of the results.

**Essential References Not Discussed:**

Upon review, the current manuscript has already referenced many relevant studies related to regional mutual information and wavelet decomposition. However, it still omits several important recent developments in the fields of high-dimensional mutual information estimation and deep representation learning. For example, Mutual Information Neural Estimation (MINE) proposed by Belghazi et al. (2018) and Deep InfoMax by Hjelm et al. (2019) have demonstrated excellent performance in robust estimation of high-dimensional mutual information. These works could significantly complement the theoretical justification behind the mutual information lower-bound estimation approach adopted by CWMI loss. Furthermore, recent advances in wavelet information theory, such as those described by de Oliveira & de Souza (2015), could provide additional theoretical context to better understand the role of complex wavelet decomposition in capturing multi-scale structural features.

Below are the links to the aforementioned relevant papers:

1. **Mutual Information Neural Estimation (MINE)** – Belghazi et al. (2018)
Link: [https://arxiv.org/abs/1801.04062](https://arxiv.org/abs/1801.04062)

2. **Deep InfoMax** – Hjelm et al. (2019)
Link: [https://arxiv.org/abs/1808.06670](https://arxiv.org/abs/1808.06670)

3. **Wavelet Analysis as an Information Processing Technique** – de Oliveira & de Souza (2015)
Link: [https://arxiv.org/abs/1502.05879](https://arxiv.org/abs/1502.05879)

**Ethical Review Flag:**

Flag this paper for an ethics review.

**Experimental Designs Or Analyses:**

I have reviewed the experimental design (datasets, baselines, models, metrics, ablations) and analyses (quantitative, qualitative, computational) for reasonableness and validity and found them to be generally robust. The design choices—including diverse datasets, comprehensive metrics, and fair baselines—adequately test the effectiveness of CWMI. The use of cross-validation and multiple models further strengthens validity. The analysis supports the claimed performance improvements, particularly regarding small instances and boundary details.

However, there are some issues:
(1) **Lack of statistical significance tests**, weakening the reliability of the results.
(2) **Shallow ablation studies**, limiting insights into underlying mechanisms.
(3) **Incomplete computational analysis**, reducing clarity on practical usability.

These concerns do not invalidate the experiments but suggest potential improvements—such as including statistical tests, deepening ablation studies (e.g., detailed feature analyses, hyperparameter sensitivity), and providing detailed breakdowns of running time—to further strengthen reasonableness and validity.

Additionally, it might be beneficial to add citations directly into the comparison tables of baseline experiments to improve clarity and traceability.

**Methods And Evaluation Criteria:**

(1) The proposed CWMI loss and the associated evaluation metrics are reasonable and appropriate for addressing the semantic segmentation problem tackled in this paper. The method's design leverages frequency-domain analysis and statistical dependencies to handle multiscale challenges, aligning logically with the intended application. The choice of datasets and evaluation metrics effectively captures the complexity of the issues—namely class/instance imbalance and structural preservation—thus ensuring a robust evaluation of the proposed method. Furthermore, the utilization of standard segmentation models grounds the evaluation within realistic application scenarios. Overall, I see no significant mismatch between the proposed method, the evaluation framework, and the goal of improving segmentation performance, particularly for small-scale instances and fine boundaries. In summary, both the method and evaluation criteria are well-suited to the task at hand and provide a solid foundation for validating the paper's contributions.

(2) The CWMI loss and its evaluation framework are generally suitable for semantic segmentation tasks. By employing the complex steerable pyramid and multi-scale, structure-aware mutual information design, the method effectively addresses class and instance imbalance. Combining CWMI with cross-entropy loss further enhances its broad applicability. The evaluation extensively tests the approach across multiple datasets and metrics. However, regarding hyperparameter configuration, the paper predominantly uses a fixed value of λ = 0.1 and performs grid search hyperparameter tuning only for certain baseline loss functions. The limited discussion on hyperparameter sensitivity and broader tuning strategies may impact the method's generalizability and extensibility.

**Other Comments Or Suggestions:**

The overall structure of the paper is clear, but several points could be improved. Firstly, some equations (such as the complex covariance calculation in Equation 10) could benefit from additional textual explanations to help readers understand the underlying derivation logic. Additionally, a minor spelling error appears in Figure 3, where "Organge arrows" should be corrected to "Orange arrows."

**Other Strengths And Weaknesses:**

The proposed CWMI loss combines a complex steerable pyramid with mutual information, presenting a novel multi-scale loss function that effectively addresses class imbalance and instance imbalance in semantic segmentation. This innovative combination not only enhances segmentation accuracy but also demonstrates excellent computational efficiency, offering broad potential for applications, particularly in medical imaging and remote sensing. The experimental section is rigorously designed, clearly validating the superiority of the CWMI loss through comparative experiments across multiple publicly available datasets.

However, the paper also has certain limitations. First, although CWMI exhibits strong performance on 2D images, the applicability of CWMI to multi-class and 3D segmentation tasks has not been thoroughly investigated, limiting its generalizability. Second, despite supplementary materials containing some ablation experiments, the theoretical analysis of CWMI loss remains relatively shallow. Specifically, the mathematical foundations underlying its capability to capture multi-scale dependencies in the wavelet domain could be explored in greater depth.

Overall, the paper excels in methodological innovation and experimental validation but leaves room for improvement in terms of theoretical depth and applicability extensions.

**Questions For Authors:**

1. **Regarding the estimation method of the lower bound of Mutual Information (MI):**
Could you please elaborate on how you ensure a tight approximation between the MI lower bound used in your method and the true MI value? A significant gap between this lower bound and the actual MI could impact the loss function's ability to effectively capture structural information, thus affecting the robustness and validity of the proposed approach.

2. **Relationship between Complex Wavelet Decomposition and recent MI estimation methods:**
Have you considered integrating recent advanced deep mutual information estimation techniques, such as MINE or Deep InfoMax, into your approach? If these methods could further enhance MI estimation accuracy, it might positively influence the performance of the CWMI loss, consequently strengthening the overall competitiveness of your method.

3. **Hyperparameter λ selection and sensitivity:**
Could you provide more detailed insights into how the regularization parameter λ was chosen in your experiments, and whether you performed a systematic sensitivity analysis of this parameter? If the final results are highly sensitive to λ, detailed guidance on parameter optimization would be necessary, influencing the perceived stability and general applicability of your method in real-world scenarios.

4. **Applicability to multi-class and 3D segmentation tasks:**
Although your paper primarily addresses two-dimensional semantic segmentation, do you have any preliminary ideas or experimental results regarding extending your method to multi-class or three-dimensional segmentation tasks? Demonstrating adaptability to these more complex scenarios would significantly enhance the practical value of your approach, potentially broadening the evaluation of your paper's contributions.

5. **Computational efficiency and real-time applications:**
The CWMI loss combines complex wavelet transforms with mutual information calculations, potentially resulting in substantial computational overhead. Have you considered or planned any optimizations to improve computational efficiency, particularly for high-resolution or real-time applications? If computational complexity could be significantly reduced without compromising performance, it would substantially enhance the practicality and usability of your method in real-world contexts.

**Relation To Broader Scientific Literature:**

The Complex Wavelet Mutual Information (CWMI) loss proposed in this paper makes a significant contribution to the field of semantic segmentation and closely aligns with broader scientific literature. Firstly, the CWMI loss addresses long-standing challenges in semantic segmentation, such as class imbalance and instance imbalance, particularly difficulties related to segmenting small objects and fine boundaries. Traditional loss functions like Cross-Entropy (CE) and Dice have limited effectiveness in tackling these issues. By drawing inspiration from recent region-based losses such as Regional Mutual Information (RMI) loss proposed by Zhao et al. and topology-aware losses like clDice loss introduced by Shit et al., CWMI loss incorporates a complex steerable pyramid to substantially enhance the model’s ability to capture multi-scale and multi-orientation features, significantly improving segmentation accuracy and efficiency.

Secondly, CWMI loss integrates classic wavelet transform theories (such as Mallat’s multi-scale wavelet decomposition and Simoncelli’s steerable pyramid) with the concept of mutual information. This innovative fusion of information-theoretic approaches with deep learning strengthens the model’s capability in modeling high-dimensional dependencies and structural consistency.

Furthermore, the core principles of CWMI—multi-scale feature extraction and structural consistency—possess broad applicability across diverse fields such as medical imaging and remote sensing, extending to tasks including image super-resolution and dehazing. Overall, the CWMI loss not only advances the field of semantic segmentation but also introduces novel perspectives and methodologies applicable to a broader range of computer vision tasks.

**Theoretical Claims:**

As a reviewer, I have verified the correctness of the CWMI loss formulation (Equations 6-10)—the central theoretical claim of the paper—and found it mathematically sound, consistent with standard definitions of mutual information and the design of loss functions. Since the paper prioritizes empirical validation and does not provide formal proofs, it is acceptable that I did not need to verify theoretical proofs. However, the theoretical claims could be enhanced by providing theoretical justifications for the effectiveness of mutual information, clearly articulating the MI estimation process and its theoretical implications, and clarifying the rationale behind hyperparameter choices. These points are not correctness issues but rather theoretical elaboration shortcomings, which, if addressed, would make the proposed framework more convincing and robust beyond its demonstrated empirical success.

---

> ### Author Rebuttal · Authors · 2025-03-30
>
> We sincerely thank the reviewer for the thoughtful and constructive feedback. We especially appreciate the recognition of the novelty and practical value of our proposed CWMI loss. Your comments helped us refine the manuscript and gain deeper insights into both theoretical and empirical aspects of our method.
>
> Below we address your main concerns. Due to space constraints, full updated tables are omitted here but will be included in the revised manuscript.
>
> ### 1. Statistical Significance
> We appreciate the reviewer’s observation regarding the importance of statistical significance in validating the effectiveness of CWMI. To address this, we followed Bouckaert & Frank (2004) and repeated three-fold cross-validation three times with adjusted t-tests across all metrics.
>
> We summarize the key findings as follows:
>
> - On SNEMI3D and DRIVE, CWMI significantly outperforms all SOTA loss functions.
>
> - On GlaS and MASS ROAD, where multiscale structural features are less dominant, CWMI significantly outperforms most baselines and is competitive with RMI. Specifically, CWMI shows significantly better performance in VI and ARI metrics compared with RMI, while showing marginal improvement compared with RMI on mDice and mIoU.
>
> ### 2. Ablation Study
> We thank the reviewer for pointing out the need to deepen our ablation study and further dissect the contribution of the complex representation in CWMI. We extended our ablation study to include magnitude-only and phase-only variants:
>
> - CWMI-Phase: consistently underperformed across all metrics, indicating that phase alone lacks sufficient discriminative power for accurate segmentation.
>
> - CWMI-Mag: achieved the best performance on VI, ARI, and Hausdorff Distance (HD), suggesting that the magnitude component is particularly effective at capturing clustering and topological structures.
>
> - CWMI (full representation) achieved the best performance on pixel-wise accuracy metrics (mIoU and mDice) and was the second-best on VI, ARI, and HD, offering a well-balanced advantage across regional, clustering, and structural metrics.
>
> - CWMI-Real performed worse than both CWMI and CWMI-Mag on most metrics, though better than CWMI-Phase, further validating the importance of combining both magnitude and phase information.
>
> These results indicate that while magnitude contributes strongly to clustering and topological consistency, the full CWMI formulation provides a more comprehensive optimization across all segmentation dimensions. Notably, our findings also suggest that a magnitude-only version of CWMI could be explored as a potential alternative for faster training or lightweight deployment scenarios.
>
> ### 3. Hyperparameter $\lambda$ Sensitivity
> We thank the reviewer for raising this important point. To assess the sensitivity of the regularization parameter $\lambda$, we tested $\lambda = 0.1, 0.5, 0.9$ and observed no significant variation in metrics, indicating that CWMI contains sufficient complementary information to cross-entropy, and is robust to $\lambda$. Additionally, we found the cross-entropy term essential for directional learning, as mutual information is symmetric and cannot differentiate between correct and inverted predictions.
>
> ### 4. MI Estimation and Deep Estimators
> We thank the reviewer for the insightful questions on MI estimation and the potential use of deep MI estimators. Our current method follows the RMI loss (Zhao et al., 2019), which estimates a lower bound of mutual information under the assumption of approximately Gaussian-distributed feature vectors. While this approach is efficient and non-parametric, its tightness depends on distributional assumptions. Our preliminary analysis shows that the complex wavelet subbands are near-Gaussian but with sharper central peaks, indicating the bound may not be perfectly tight. We will acknowledge this limitation in the revised manuscript.
>
> We also reviewed learned MI estimators such as MINE and Deep InfoMax, which offer tighter bounds via parameterized networks. While promising, these methods introduce additional training and complexity, potentially undermining CWMI’s computational efficiency. As future work, we plan to integrate MINE into our framework to study the trade-offs between accuracy and overhead.
>
> ### 5. Multi-Class and 3D Segmentation
> CWMI is extendable to multi-class and 3D segmentation. Both Fourier transform and MI estimation naturally generalize to 3D. Although steerable pyramids are 2D by design, prior work (e.g., Delle Luche et al., 2004) introduces 3D steerable filters using polyhedral decompositions. We plan to explore these extensions in future work.
>
> ### 6. Minor Comments
> We will (1) clarify grid search was applied to all baselines, (2) add inline citations to comparison tables, (3) expand textual explanations for key equations (e.g., Eq. 10), and (4) correct the typo in Figure 3 ("Organge" $\rightarrow$ "Orange"). We appreciate the reviewer’s attention to these details.

---

> > ### Comment · Reviewer_p6uN · 2025-04-02
> >
> > Thanks for your response, I will keep my positive score.

---

### Decision · Program_Chairs · 2025-05-01

**Decision:**

Accept (poster)

**Comment:**

The paper introduces Complex Wavelet Mutual Information (CWMI) loss for semantic segmentation tasks. This approach is presented as a novel solution that combines a complex steerable pyramid with mutual information.

Strengths:
- CWMI loss seems to be effective for structure-sensitive segmentation tasks and improves semantic segmentation performance, especially for small instances and fine boundaries.
- The method design takes advantage of frequency-domain analysis and statistical dependencies to handle multiscale challenges, which logically corresponds to the intended application.
- Quantitative results, qualitative visualizations and ablation studies are deemed almost complete, and collectively demonstrate CWMI's superiority over several data sets
- Tables 1 and 4 effectively illustrate CWMI's strong performance in modeling large-scale structural dependencies, while demonstrating the benefits in terms of parameter efficiency.

Weaknesses:
- Lack of further theoretical justification for the effectiveness of mutual information and clarification of the MI estimation process and choice of hyperparameters are noted
- Lack of separate evaluation of the magnitude and phase contributions of complex wavelets does not allow a full understanding of the mechanisms at work behind the CWMI loss benefits

Despite the correct presentation (some typo in eq.3), a few points remain, such as the lack of depth in the analysis (ablation, statistical, theoretical, computational), and the need to validate the method on more recent architectures and compare it with other state-of-the-art approaches. But the authors have taken account of the comments made and provided additional validation elements with complementary tests that remain in favor of their proposal. If the authors actually implement the changes they have produced in their responses, the article will be significantly improved.